# Chiral metal-organic frameworks incorporating nanozymes as neuroinflammation inhibitors for managing Parkinson's disease

Wei Jiang [1,2,5], Qing Li [1,5] ✉, Ruofei Zhang[3], Jianru Li[3], Qianyu Lin[1], Jingyun Li[2,3], Xinyao Zhou[4], Xiyun Yan [2,3] ✉ & Kelong Fan [2,3] ✉

Nanomedicine-based anti-neuroinflammation strategy has become a promising dawn of Parkinson's disease (PD) treatment. However, there are significant gaps in our understanding of the therapeutic mechanisms of antioxidant nanomedicines concerning the pathways traversing the blood-brain barrier (BBB) and subsequent inflammation mitigation. Here, we report nanozyme-integrated metal-organic frameworks with excellent antioxidant activity and chiral-dependent BBB transendocytosis as anti-neuroinflammatory agents for the treatment of PD. These chiral nanozymes are synthesized by embedding ultra-small platinum nanozymes (Ptzymes) into L-chiral and D-chiral imidazolate zeolite frameworks (Ptzyme@L-ZIF and Ptzyme@D-ZIF). Compared to Ptzyme@L-ZIF, Ptzyme@D-ZIF shows higher accumulation in the brains of male PD mouse models due to longer plasma residence time and more pathways to traverse BBB, including clathrin-mediated and caveolae-mediated endocytosis. These factors contribute to the superior therapeutic efficacy of Ptzyme@D-ZIF in reducing behavioral disorders and pathological changes. Bioinformatics and biochemical analyses suggest that Ptzyme@D-ZIF inhibits neuroinflammation-induced apoptosis and ferroptosis in damaged neurons. The research uncovers the biodistribution, metabolic variances, and therapeutic outcomes of nanozymes-integrated chiral ZIF platforms, providing possibilities for devising anti-PD drugs.

Parkinson's disease (PD) is a chronic, age-related neurodegenerative disorder characterized by motor dysfunction and memory impairment resulting from neuronal damage and dopamine (DA) loss in the substantia nigra pars compacta (SNpc)[1,2]. Currently, the therapeutic strategies for PD focus primarily on alternative DA-based therapies, including levodopa (L-dopa), DA agonists, and monoamine oxidase B (MAOB) inhibitors[3]. Unfortunately, despite vigorous reports, few

methods effectively inhibit the progression of PD, underlining the importance of developing novel strategies[4,5].

Previous studies have shown that the pathogenesis of PD is closely related to neuroinflammation, which is primarily caused by oxidative damage. Therefore, antioxidants and anti-inflammatory agents are promising candidates in clinical practice for treating PD[6,7]. In healthy cells, excess reactive oxygen species (ROS) and reactive

[1]Application Center for Precision Medicine, the Second Affiliated Hospital of Zhengzhou University, Henan 450052, China. [2]Nanozyme Medical Center, Academy of Medical Sciences, Zhengzhou University, Zhengzhou 450001, China. [3]CAS Engineering Laboratory for Nanozyme, Key Laboratory of Biomacromolecules, Institute of Biophysics, Chinese Academy of Sciences, 15 Datun Road, Beijing 100101, China. [4]School of Engineering and Applied Science, University of Pennsylvania, Philadelphia 19104, USA. [5]These authors contributed equally: Wei Jiang, Qing Li. ✉e-mail: lq1515012032@163.com; yanxy@ibp.ac.cn; fankelong@ibp.ac.cn

nitrogen species (RNS) are regulated by the antioxidant defense system, including superoxide dismutase (SOD), catalase (CAT), glutathione peroxidase (GPX), low molecular weight antioxidants (glutathione and vitamins C and E)[8] and other antioxidant defense systems. However, therapies using free enzymes often fail to meet the clinical treatment demands due to their shortcomings, such as easy inactivation, short half-life, and limited membrane permeability[9]. Therefore, designing artificial enzyme systems with enhanced antioxidant activity, improved stability, prolonged circulation time, and enhanced transmembrane transport capability poses a challenging yet crucial avenue of research.

Recent advances in nanozymes have prompted the rapid development of nanomaterial-mediated antioxidant therapies[10–12]. Nanozymes possess the advantages of lower cost and more straightforward preparation techniques compared to natural enzymes[13]. In addition, nanozymes exhibit superior stability and membrane permeability compared to natural enzymes and present great potential in the biomedical field. A growing number of nanozymes have been reported to exhibit satisfactory enzyme-mimetic activities due to their small size and structure-dependent function, and have been used in bioanalysis, environmental monitoring, and disease therapy[14,15]. For example, platinum nanozymes (Ptzymes) have been reported to exhibit SOD-, CAT- and uricase-like activities[16,17], and have been applied in biomedical research, such as tumor catalytic therapy[18,19]. However, clinical applications of stand-alone nanozyme therapies have also been noted to be problematic due to short vascular circulation times and susceptibility to inter-particle aggregation[20]. Fortunately, the use of nanozyme in zeolite imidazolate framework (ZIF) has been reported to be a promising approach for developing nanozyme therapies[21,22]. Being a low-toxicity and biocompatible metal-organic framework (MOF), ZIF remains stable in physiological environments while exhibiting degradation in acidic environments, including tumor microenvironments (TME) and inflammatory sites[21]. Moreover, the properties of nanoparticles (NPs) encapsulated in ZIF can be further altered to prevent NPs aggregation[22]. Therefore, encapsulation with ZIF may improve the blood residence time, transmembrane transport, and cellular uptake of NPs. Moreover, Jiang et al. have demonstrated that the biological toxicity, pharmacokinetics and in vivo biodistribution of NPs are efficiently adjusted by modification of chirality ligands on their surface[23]. Qu et al. have found that ligands with different chiral structures endow NPs with different substrate selectivity and various biological functions[24,25]. However, comprehensive mechanistic investigations on the influence of chiral structures on neurodegenerative disease treatments remain markedly insufficient. Further elucidation is required regarding the diverse chiral mechanisms of action, in vivo transport mechanisms, and therapeutic effects. This research endeavor aims to offer valuable insights and guidance for the effective implementation of chiral drugs in clinical settings.

Here, we rationally designed nanozyme-integrated chiral MOFs by embedding Ptzymes in L- and D-type chiral ZIF shells via shell-ligand exchange reactions, and named them Ptzyme@L-ZIF and Ptzyme@D-ZIF, respectively. Chemical characterization of the nanozyme-integrated chiral MOFs proved that they successfully load Ptzymes and exhibit a typical ZIF crystal structure. Analysis of enzymatic activity suggested that both Ptzyme@L-ZIF and Ptzyme@D-ZIF exhibit excellent cascade SOD- and CAT-mimetic activities. Real-time living cell tracer imaging showed that Ptzyme@D-ZIFs achieve transcytosis by clathrin-mediated and caveolae-mediated endocytosis, benefiting their BBB traverse effect. Moreover, analysis of blood clearance and tissue distribution showed that Ptzyme@D-ZIFs are more stable in plasma and exhibit superior accumulation in the brain compared to Ptzyme@L-ZIFs, with more satisfactory therapeutic effects in the subsequent behavioral and pathological assessments on PD mouse models. Furthermore, transcriptome analysis and in vitro studies confirmed

that the therapeutic mechanism of Ptzyme@D-ZIFs for PD symptoms is largely related to the mitigation of neuronal apoptosis and ferroptosis, which are induced by excessive ROS and dysregulated neuroinflammation during PD disease progression (Fig. 1).

## Results and discussion

### Synthesis and characterization of Ptzyme-integrated chiral ZIFs

The composition of Ptzyme@ZIFs is shown in Fig. 1. Initially, Ptzymes were synthesized by in-situ growth of Pt nanozymes on polyvinylpyrrolidone (PVP) polymer with varying molecular weights (K12, K16, K30, and K90). The SOD- and CAT-like activities of the four Ptzyme variants with different degrees of crystallization were shown in Fig. 2A, Supplementary Fig. 1, and Supplementary Fig. 2, and it was suggested that the Ptzyme synthesized using K30 as a template exhibit suitable activity compared to other templates. Based on these results, K30 was selected as the template for the subsequent experiments. The synthesized Ptzymes using K30 as the template exhibited an ultrasmall size, with diameters ranging from 6 to 8 nm, as characterized by transmission electron microscopy (TEM) (Fig. 2B). Subsequently, Ptzyme@ZIFs were prepared by encapsulating the Ptzymes in ZIF by biomimetic mineralization. Finally, chiral amino acids (L-histidine and D-histidine) were incorporated into Ptzyme@ZIFs through the competitive coordination of the imidazole groups on histidine (His) with $Zn^{2+}$ and a post-synthetic ligand exchange to obtain chiral Ptzyme@ZIF (Ptzyme@L-ZIF and Ptzyme@D-ZIF). As shown in Fig. 2C-E, Ptzyme@ZIFs exhibited uniform cube morphologies with a monodisperse of ~200 nm. For His-modulated synthesis, Ptzyme@L-ZIFs and Ptzyme@D-ZIFs (Fig. 2F) showed similar morphologies. However, the particle size of ZIF shells decreased to sub-90 nm in diameter after incorporated with chiral His, as characterized by HRTEM. Meanwhile, Ptzyme@L-ZIFs and Ptzyme@D-ZIFs exhibited similar particle sizes and structures, suggesting that the modification with L-His and D-His do not induce any apparent morphological changes at the microscopic level. The X-ray diffraction (XRD) pattern of Ptzyme was in accordance with the cubic phase of Pt with a {111} and {200} facets (Supplementary Fig. 1), which is consistent with 0.2 nm of lattice spacing characterized by TEM (Fig. 2G). Dynamic light scattering (DLS) analysis revealed that Ptzyme@ZIFs, Ptzyme@L-ZIFs, and Ptzyme@D-ZIFs exhibited hydrodynamic particle sizes of $190.35 \pm 48.16$ nm, $82.24 \pm 17.33$ nm, and $70.37 \pm 15.84$ nm, respectively, indicating that His-modulated strategy affects the size distribution of Ptzyme@ZIFs (Fig. 2H). Moreover, the average diameter of Ptzyme@D-ZIFs in deionized water remained relatively unchanged during 5 days of observation (Supplementary Fig. 4), which demonstrates the considerable stability of the Ptzyme@D-ZIFs. In addition, His incorporation reduced the mean zeta potential of Ptzyme@ZIFs from ~+29.17 mV to ~+18.4 mV of Ptzyme@D-ZIF and +25.05 mV of Ptzyme@L-ZIF, respectively (Fig. 2I). These alterations in zeta potential further demonstrate the effect of the His-modulated strategy on the surface charge characteristics of the Ptzymes. Furthermore, the elemental mapping of Ptzyme@D-ZIFs further verified the presence of Pt signatures within the composite, suggesting the incorporation of Ptzymes in ZIF shells (Fig. 2J). In addition, as shown in Fig. 2K, the powder X-ray diffraction (PXRD) patterns of Ptzymes, Ptzyme@ZIFs, Ptzyme@L-ZIFs, and Ptzyme@D-ZIFs showed peaks that were in good agreement with the simulated ZIF-8. These results suggest that ZIF-8 frameworks are formed in the biomimetic mineralization, and the structural integrity of ZIF-8 is retained in the presence of Ptzymes and amino acid modifications. Moreover, the crystal form of embedded Ptzyme is preserved and consistent with the selected area electron diffraction (SAED) image (Supplementary Fig. 3). Furthermore, the circular dichroism (CD) spectra of Ptzyme@ZIF, Ptzyme@L-ZIF, and Ptzyme@D-ZIF (Fig. 2L) showed a specular CD peak in the region below 216 nm, further confirming the successful modification of surface ligands with opposite chirality. The same properties are shown

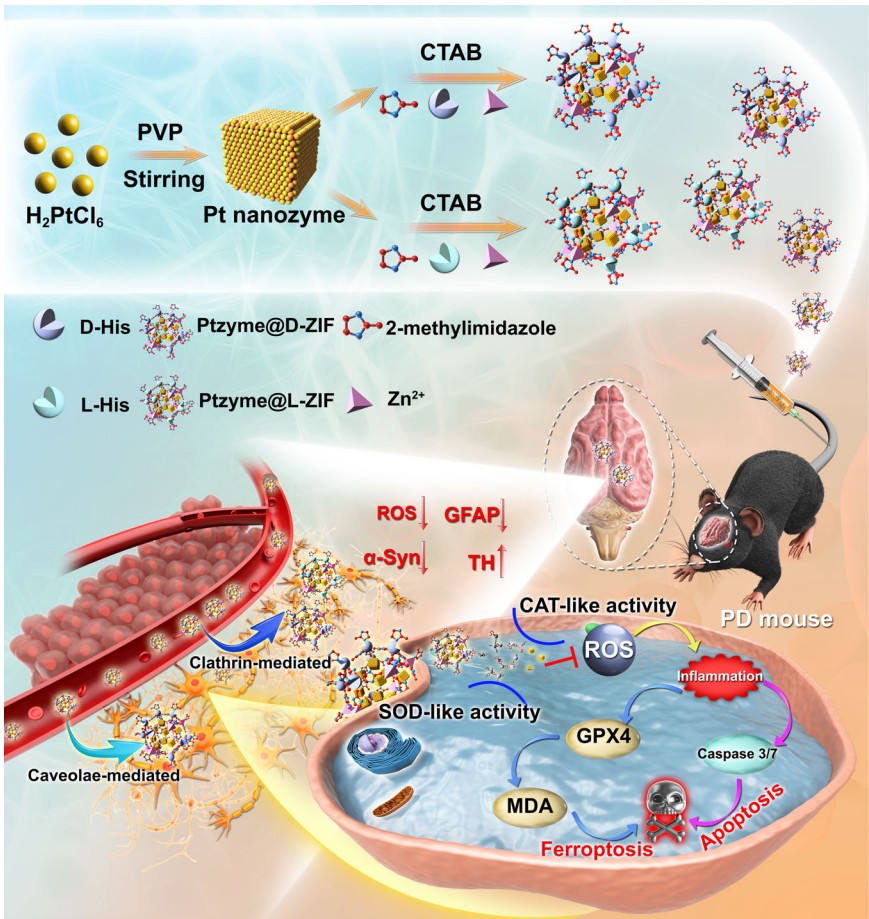

**Fig. 1 | Schematic illustration of Ptzymes integrated L- and D-chiral ZIFs and their therapeutic mechanism on PD based on remission of both apoptosis and ferroptosis on neurons injured by excessively produced ROS and disordered inflammation.** PVP polyvinylpyrrolidone, CTAB hexadecyl trimethyl ammonium bromide, GFAP glial fibrillary acidic protein, TH: tyrosine hydroxylase, GPX4 glutathione peroxidase 4, MDA malondialdehyde.

with other chiral amino acids (Supplementary Fig. 5). The Fourier transform infrared spectroscopy (FTIR) studies were conducted to analyze the chemical composition of the samples. In the FTIR spectra, a newly observed absorption band appeared at around 1685 cm$^{-1}$ (gray band in Fig. 2M). This absorption band was assigned to the asymmetric stretching of the carboxyl group from His. Of note, this band overlapped with another broad band at around 1600 cm$^{-1}$. The broad band at 1600 cm$^{-1}$ was assigned to the overlapping of vibrations, namely the C = N stretching of the imidazole group derived from His and 2-Methylimidazole (HmIM, round 1580 cm$^{-1}$). Taken together, the appearance of the absorption band in FTIR spectra at 1685 cm$^{-1}$ indicates the presence of the carboxyl group from His, while the overlapping band at 1600 cm$^{-1}$ suggests the presence of C = N stretching vibrations from the imidazole group and HmIM. These results indicate that Ptzyme@L-ZIFs and Ptzyme@D-ZIFs are successfully synthesized.

Next, the enzyme-like activities of Ptzyme@ZIFs, Ptzyme@L-ZIFs, and Ptzyme@D-ZIFs were analyzed under physiological conditions. As shown in Fig. 3A, B, Supplementary Fig. 6A, and Supplementary Fig. 6B, Ptzyme@ZIFs, Ptzyme@L-ZIFs, and Ptzyme@D-ZIFs all showed SOD- and CAT-like activities in phosphate buffer (pH 7.4) and presented ROS/RNS scavenging ability. However, all of them did not show remarkable oxidase- or peroxidase-like activity under these conditions, which typically generate ROS (Supplementary Fig. 6C and Supplementary Fig. 6D). Moreover, Ptzymes integrated with different amino acid-modified ZIFs also exhibited the ability to scavenge superoxide radicals (O$_2^{\cdot-}$) (Fig. 3C). Furthermore, as detected by electron spin resonance (ESR) spectroscopy, Ptzyme@L-ZIFs and Ptzyme@D-ZIFs significantly scavenged the hydroxyl radicals (·OH)

generated by Fenton reaction (Fig. 3D) and O$_2^{\cdot-}$ produced by the xanthine-xanthine oxidase system (Fig. 3E). In addition, 2,2′-azinobis-(3-ethylbenzothiazoline-6-sulfonate) (ABTS) free radical (ABTS$^{\cdot+}$) generated from the ABTS/K$_2$S$_2$O$_8$ system was also significantly scavenged by the cascade SOD/CAT-like activities of Ptzyme@ZIFs, Ptzyme@L-ZIFs, and Ptzyme@D-ZIFs (Fig. 3F, G). It is noteworthy that the nanozyme-integrated chiral ZIFs have significantly improved RNS scavenging capacity compared to the achiral ones (Ptzyme@ZIFs), which is probably due to their superior dispersibility caused by amino acid modification. Ptzyme@D-ZIFs exhibited slightly higher SOD and CAT-like activities, along with significantly enhanced RNS clearing capacity compared to Ptzyme@L-ZIFs. These improvements may be attributed to variations in ligand arrangement and electronic environment. This is substantiated by the variations in binding charge between chiral nanozymes and substrate (Supplementary Fig. 7). Moreover, factors like active site exposure and conformational effects, steming from distinct chiral ligand modifications, may further contribute to the superior catalytic performance of Ptzyme@D-ZIFs[24,26,27]. These results indicate that Ptzyme@L-ZIFs and Ptzyme@D-ZIFs successfully mimic the cascade SOD/CAT activity of natural enzymes and exhibit great potential to alleviate PD as antioxidant agents.

## BBB traverse capacity and related mechanisms

As the most crucial barrier between blood and brain parenchyma, the BBB possesses essential physiological functions for the normal state of the central nervous system (CNS). It selectively prevents the entry of toxic substances into the brain, thus simultaneously inhibiting the uptake of functional molecules in PD therapy[28,29]. For example, the

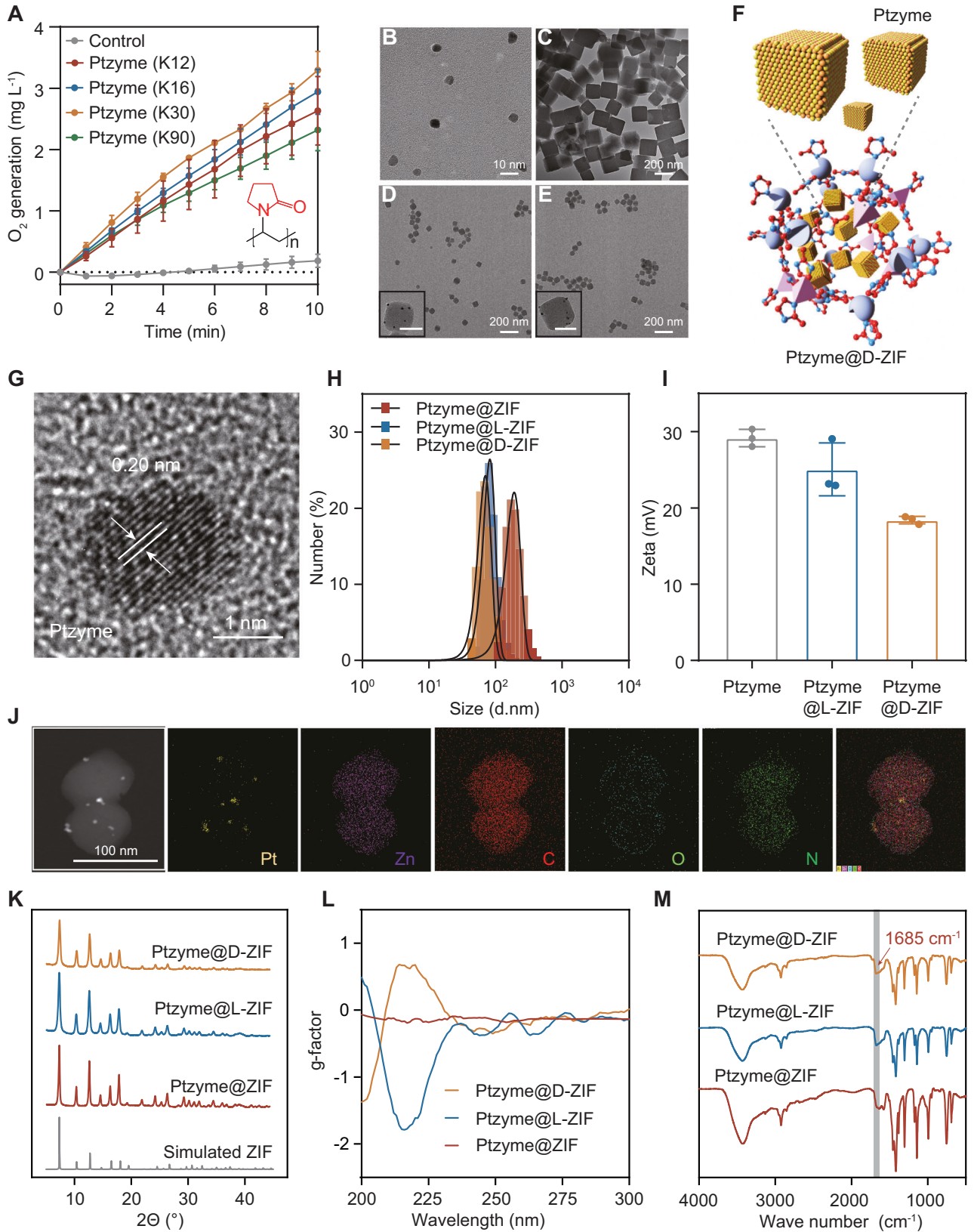

therapeutic potential of vitamin B12 in PD is limited due to its restricted ability to traverse the BBB and its lower binding rate with transcobalamin II [30]. The bEnd.3 cell line, derived from brain microvascular endothelial cells, has the capability to form dense cellular layers with tight junctions. These characteristics make bEnd.3 cells a commonly employed model for in vitro simulation of the BBB[29]. In light

of this, our initial objective involved assessing the ability of nanozyme-integrated chiral ZIFs to traverse the BBB and be taken up by neuronal cells in an in vitro BBB model. This model involved the co-culture of bEnd.3 cells with human neuroblastoma cells (SH-SY5Y) in a transwell system (Fig. 4A). As shown in the fluorescence micrograph of SH-SY5Y cells in the lower chamber, FITC-Ptzyme@D-ZIFs efficiently passed

**Fig. 2 | Structural characterization of Ptzyme integrated chiral ZIFs. A** The CAT-like activities of Ptzymes using PVP with different molecular weights (K12, K16, K30, and K90), $n = 3$ independent experiments, Data represent the mean ± SD. TEM images of (**B**) Ptzyme, (**C**) Ptzyme@ZIF, (**D**) Ptzyme@L-ZIF (insert is TEM image of Ptzyme@L-ZIF; scale bar is 50 nm), and (**E**) Ptzyme@D-ZIF (insert is TEM image of Ptzyme@D-ZIF; scale bar is 50 nm). **F** Schematic illustration of of Ptzyme and Ptzyme@D-ZIF. **G** High-resolution TEM (HR-TEM) image of Ptzymes embedded in ZIF shell. **H** Diameter of Ptzyme@ZIF, Ptzyme@L-ZIF, and Ptzyme@D-ZIF in deionized water. **I** Zeta potential of Ptzyme@ZIF, Ptzyme@L-ZIF, and Ptzyme@D-ZIF in deionized water, $n = 3$ independent experiments. Data represent the mean ± SD. **J** Element distribution of Ptzyme@D-ZIF. **K** PXRD patterns of simulated ZIF-8, Ptzyme@ZIF, Ptzyme@L-ZIF, and Ptzyme@D-ZIF. **L** CD spectra of Ptzyme@ZIF, Ptzyme@L-ZIF, and Ptzyme@D-ZIF. **M** FT-IR spectra of Ptzyme@ZIF, Ptzyme@L-ZIF, and Ptzyme@D-ZIF.

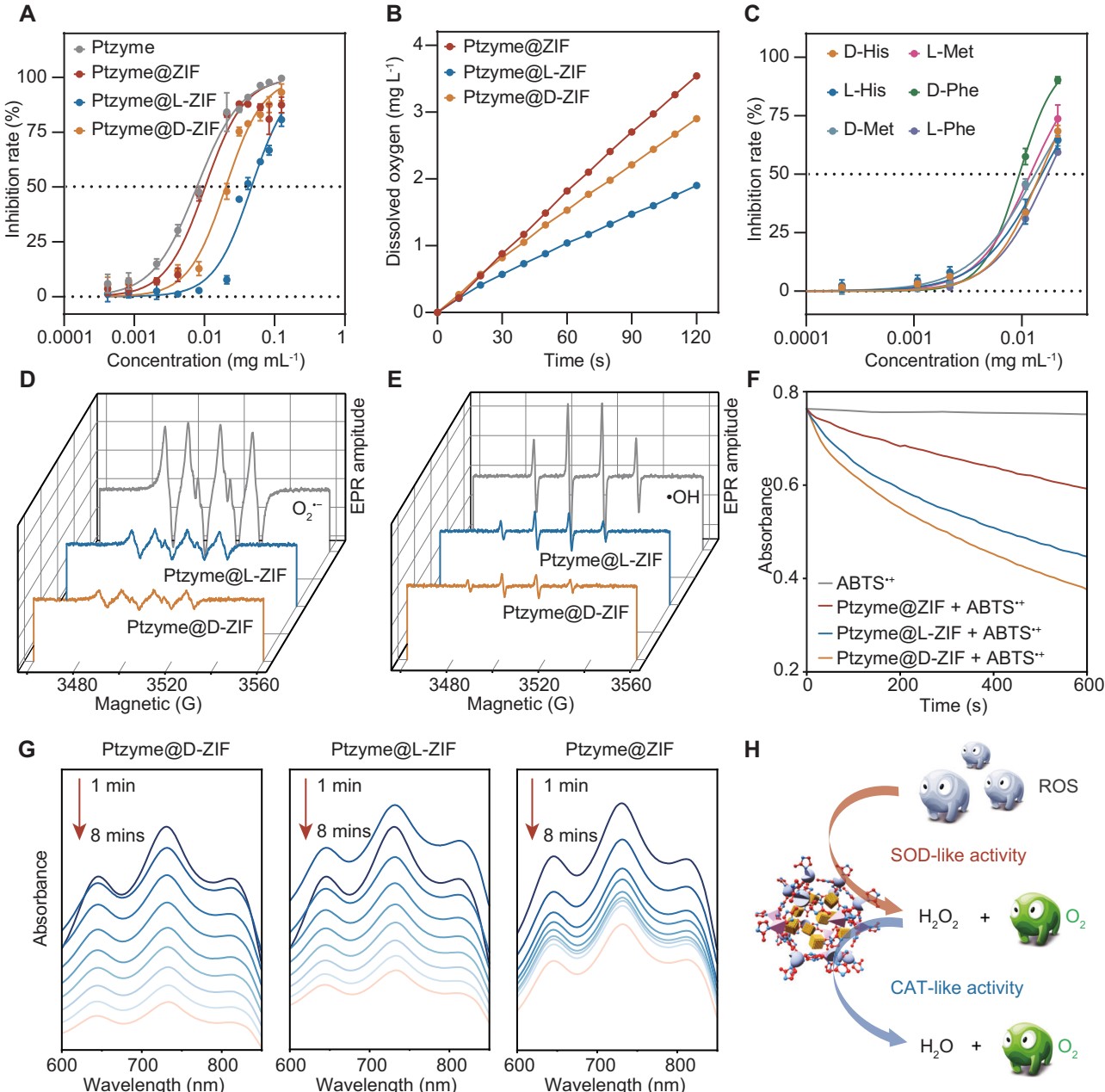

**Fig. 3 | ROS scavenging by Ptzyme integrated chiral ZIFs. A** The SOD-like activities of Ptzyme, Ptzyme@ZIF, Ptzyme@L-ZIF, and Ptzyme@D-ZIF, $n = 3$ independent experiments. Data represent the mean ± SD. **B** The CAT-like activities of Ptzyme@ZIF, Ptzyme@L-ZIF, and Ptzyme@D-ZIF. **C** The SOD-like activities of amine acid modified Ptzyme@ZIF, $n = 3$ independent experiments. Data represent the mean ± SD. **D, E** Ptzyme@D-ZIF nanozymes reduce the generation of superoxide radicals and hydroxyl radicals illustrated by ESR spectroscopy. **F** ABTS[·+] spectra at 734 nm were recorded from 0 to 600 s after the addition of Ptzyme@ZIF, Ptzyme@L-ZIF, and Ptzyme@D-ZIF. **G** UV-vis spectra were recorded at 1 - 8 mins after the addition of Ptzyme@ZIF, Ptzyme@L-ZIF, and Ptzyme@D-ZIF. **H** Schematic illustration of nanozyme-integrated chiral ZIFs-mediated antioxidative activities.

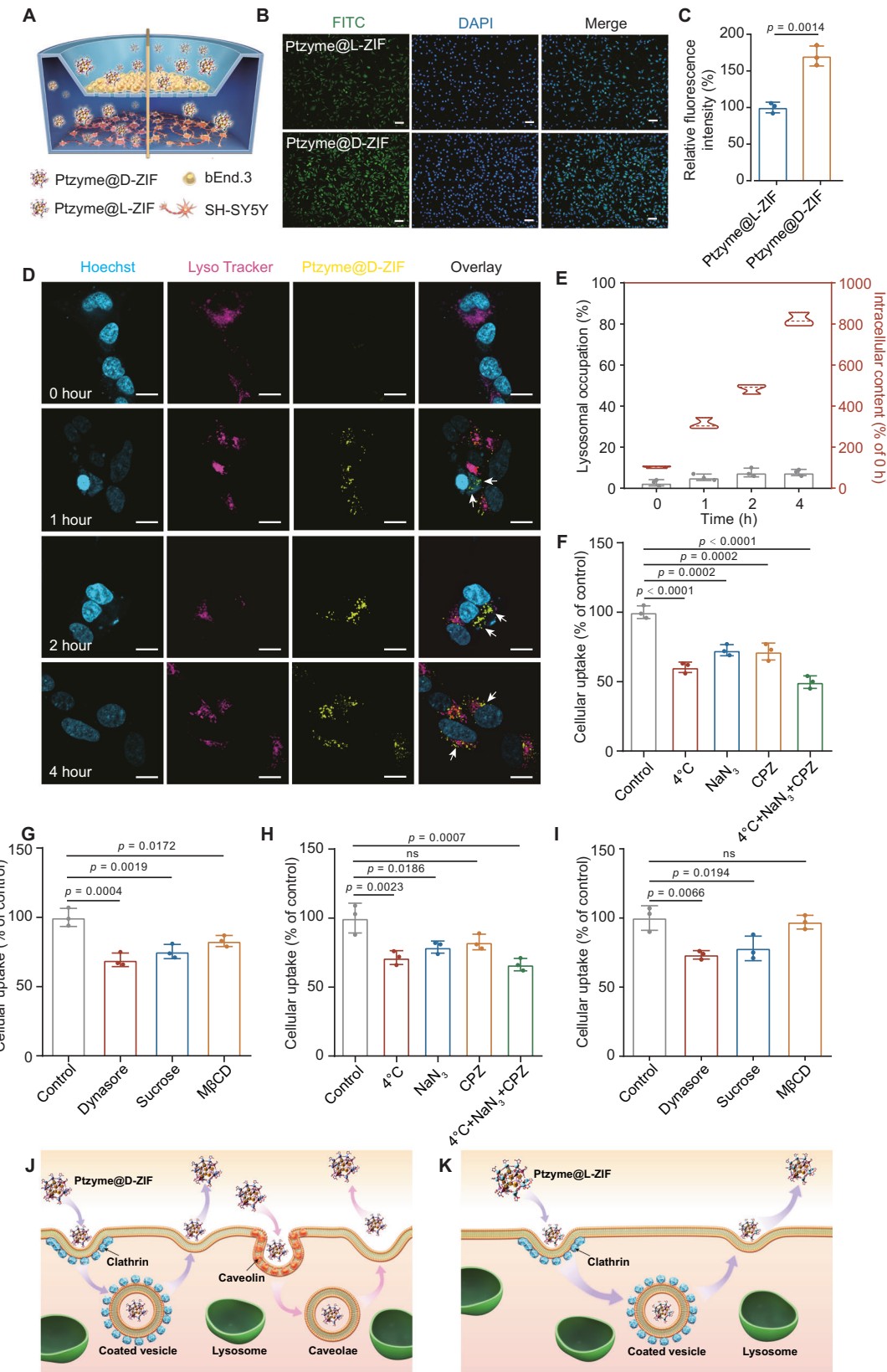

through the in vitro BBB model and were endocytosed into SH-SY5Y cells after co-cultured for 24 h (Fig. 4B). The fluorescence intensity of SH-SY5Y cells treated with FITC-Ptzyme@D-ZIFs was significantly higher than that of their L-counterparts (Fig. 4C), suggesting that compared with Ptzyme@L-ZIFs, Ptzyme@D-ZIFs exhibit more effective BBB traverse ability, and they are more easily endocytosed by neurons. In addition, Ptzyme@ZIFs without labeling fluorescent dyes were also used to investigate their ability to traverse the BBB. TEM images showed that Ptzyme@D-ZIFs successfully entered the SH-SY5Y cells, while negligible amounts of Ptzyme@L-ZIFs entered into cells (Supplementary Fig. 8), suggesting the higher permeability of Ptzyme@D-ZIF traversing the BBB and the superior endocytosis by

**Fig. 4 | BBB traversing ability and relevant mechanisms of Ptzyme integrated chiral ZIFs. A** Schematic demonstration of the BBB pattern using transwell assay, (**B**) the fluorescence microscope images, and (**C**) the corresponding quantitative results of FITC-labeled Ptzyme integrated chiral ZIFs in the lower chamber of SH-SY5Y cells. The scale bars are 50 μm, $n = 3$ independent experiments. Data represent the mean ± SD. A representative image of three biologically independent samples from each group is shown. **D** Intracellular tracking of FITC-Ptzyme@D-ZIFs in bEnd.3 cells after staining cells with Lyso Tracker and Hoechst. The white arrows indicate the intracellular position of Ptzyme@D-ZIFs. The scale bars are 10 μm. A representative image of three biologically independent samples from each group is shown. **E** Lysosomal occupation detected by the ratio of fluorescence overlapped

by lysosomes and Ptzyme@D-ZIFs, as well as the fluorescence intensity of bEnd.3 cells after treatment with FITC-Ptzyme@D-ZIFs, $n = 3$ independent experiments. Data represent the mean ± SD. Internalization of (**F**, **G**) Ptzyme@D-ZIFs and (**H**, **I**) Ptzyme@L-ZIFs by bEnd.3 cells under treatment of various endocytosis inhibitors, $n = 3$ independent experiments. Data represent the mean ± SD. In control group, bEnd.3 cells were incubated with nanocomposites but without endocytosis inhibitors. Schematic illustrating the cellular endocytosis and transcytosis process of (**J**) Ptzyme@D-ZIFs and (**K**) Ptzyme@L-ZIFs. The statistical analyses were conducted using GraphPad Prism 8.0.2. The outcomes were compared via one-way ANOVA (with Tukey's post hoc correction for multiple comparisons). "ns" indicates not significant.

nerve cells. Moreover, the uptake of Ptzyme@L-ZIFs and Ptzyme@D-ZIFs into the brain endothelial cells in vivo was also studied. Intercellular cell adhesion molecule-1 (ICAM-1) was used to label the inflammatory brain endothelial vessels in PD disease models, as 1-methyl-4-phenyl-1,2,3,6-tetrahy-dropyridine (MPTP) has been reported to cause an inflammatory response in the brain[2]. The result demonstrated that following treatment with Ptzyme@D-ZIFs, a notable accumulation of nanoparticles occurred in the brain endothelial vessels damaged by inflammation. In contrast, the enrichment in endothelial vessels and retention of brain tissue observed with Ptzyme@L-ZIFs were significantly lower than those with Ptzyme@D-ZIFs, thereby confirming the superior in vivo BBB penetrating ability of Ptzyme@D-ZIFs (Supplementary Fig. 9).

Subsequently, the distribution of nanozyme-integrated chiral ZIFs in the SNpc and striatum (ST) was examined using sliced brain tissues. As depicted in the brain slices (Supplementary Fig. 10A) and supported by the quantitative analysis (Supplementary Fig. 10B), a substantial fluorescence signal was detected in the SNpc and ST regions following treatment with FITC-Ptzyme@D-ZIFs. In contrast, the fluorescence signal in these regions was negligible after treatment with FITC-Ptzyme@L-ZIFs. In addition, the exact distribution of nanozyme-integrated chiral ZIFs in the glial cells was determined. As shown in Supplementary Fig. 11, the co-localization of FITC-Ptzyme@D-ZIFs fluorescence with ionized calcium binding adapter molecule-1 (Iba-1) fluorescence implies that FITC-Ptzyme@D-ZIFs efficiently enrich in glial cells. The above results comprehensively confirmed the superior BBB traverse and brain enrichment ability of Ptzyme@D-ZIFs over Ptzyme@L-ZIFs at the tissue and cell levels.

Since there was little difference in particle size between Ptzyme@D-ZIF and Ptzyme@L-ZIF, we further investigated the mechanism by which these chiral nanozyme-integrated ZIFs traverse the BBB at the cellular level. First, intracellular trafficking was assessed by FITC-labeled Ptzyme@D-ZIFs after staining lysosomes and nuclei of bEnd.3 cells with Lysotracker (red) and Hoechst (blue), respectively. The results showed that FITC-Ptzyme@D-ZIFs (green fluorescence) began to enter the bEnd.3 cells after 1 h, then gradually increased, achieved significant intracellular enrichment after 4 h (Fig. 4D, E). These results demonstrated the favorable cellular uptake of Ptzyme@D-ZIFs on BBB endothelial cells. Importantly, a little overlap of lysosomal and Ptzyme@D-ZIFs fluorescence was detected, indicating that the lysosomal-mediated pathway is not the primary endocytosis mechanism for Ptzyme@D-ZIFs (Fig. 4D, E). Previous studies have demonstrated that the lysosome-independent endocytosis pathway on BBB endothelial cells is beneficial for the transcytosis ability of nanocarriers[17,31], and this may be the reason for the promising BBB traverse ability of Ptzyme@D-ZIFs.

Furthermore, the exact uptake pathways of the nanozyme-integrated chiral ZIFs were analyzed by chemical endocytosis inhibitors. As shown in Fig. 4F, G, the cellular uptake of Ptzyme@D-ZIFs and its L-counterpart was significantly inhibited by low temperature and energy-dependent endocytosis inhibitors, including sodium azide (NaN$_3$) and chlorpromazine (CPZ). After the combined treatment with the above three inhibitors, the endocytosis process was dramatically

inhibited, suggesting that the uptake of the two nanosystems was energy-mediated. However, the cellular uptake of achiral Ptzyme@ZIF was not inhibited by low temperature and energy-dependent endocytosis inhibitors (Supplementary Fig. 12). Further exploration of specific energy-dependent endocytosis pathways revealed discrepancies between the two chiral nanosystems. Dynasore is an inhibitor of dynamin-mediated endocytosis, including both clathrin-mediated and caveolae-mediated endocytosis. Moreover, sucrose and methyl-β-cyclodextrin (MβCD) are inhibitors of clathrin-mediated and caveolae-mediated endocytosis, respectively. As shown in Fig. 4H, I, dynasore, sucrose, and MβCD all significantly inhibited the endocytosis of the Ptzyme@D-ZIFs on BBB endothelial cells. In contrast, only dynasore and sucrose reduced the cellular uptake of Ptzyme@L-ZIFs. These results indicate that Ptzyme@D-ZIFs enter BBB endothelial cells through both clathrin-mediated and caveolae-mediated endocytosis pathways, while Ptzyme@L-ZIFs only employ the clathrin-mediated endocytosis (Fig. 4J, K). Taken together, the effective lysosome-independent endocytosis pathway promotes the BBB traverse ability of Ptzyme@D-ZIFs.

In recent years, scientists have paid more and more attention to the great effect of surface modifications on the stability and biocompatibility of nanomaterials[32]. Remarkably, recent studies have highlighted the significant impact of ligand chirality in nanoparticles on biological responses to exogenous stimulations[33]. In addition, it affects the pharmacokinetics and metabolism of drugs by adjusting the stereospecific interactions between chiral components and biological entities[23,34]. In our study, we were surprised to find that different chiral His ligands also affect the cellular uptake pathways of nanoparticles. Because the experiments demonstrated that the presence or absence of His ligand and its chirality significantly altered the way nanoparticles traverse BBB, we speculated that the chirality of His and the resulting chirality of nanoparticles played an irreplaceable role in this process. Therefore, the different endocytic pathways between Ptzyme@D-ZIFs and Ptzyme@L-ZIFs could be ascribed to the two conceivable reasons: (1) D/L His ligands may lead to different cell recognition and internalization pathways; (2) The chiral structures of the nanozyme-integrated ZIFs may be distinguished by the assembly of clathrin or caveolae, leading to different cell internalization pathways.

## Blood clearance rate and brain tissue distribution

To evaluate the in vivo therapeutic potential of Ptzyme@D-ZIFs, we generated 1-methyl-4-phenyl-1,2,3,6-tetrahydropyridine (MPTP)-induced PD phenotype mouse models. The blood clearance kinetics of Ptzyme@D-ZIFs and Ptzyme@L-ZIFs were then examined. Here, nanozyme-integrated chiral ZIFs were administered intravenously to C57BL/6 mice and determined by enzymatic activity in plasma after a single injection. As shown in Supplementary Fig. 13, the plasma concentration of Ptzyme@L-ZIFs decreased to less than 20% of the initial value at 10 h after administration, with an elimination half-life of 0.64 h. In contrast, Ptzyme@D-ZIFs significantly prolonged the systemic circulation time and extended their elimination half-life to 6.72 h. This extension in blood residence time verified the improvement of their stability in vivo, which favors therapeutic efficacy in PD

models. The observed phenomenon in our study, where different chiral ligand modifications have different effects on pharmacokinetics, may be attributed to the formation of protein corona and different types of protein corona composition. These factors can affect the efficiency of nanozymes uptake by different cells, subsequently influencing the efficiency with which chiral nanozymes penetrate the BBB and are cleared by the immune clearance system in the blood[35,36].

To analyze the biodistribution of chiral nanozymes in the brain, major organs of nanozyme-integrated chiral ZIFs treated PD mice were obtained and subjected to biological transmission electron microscopy (TEM) and inductively coupled plasma mass spectrometry (ICP-MS) analysis. As shown in the TEM images (Fig. 5A), after sectioning the brain tissue into ultrathin sections, Ptzyme@D-ZIFs with cube morphologies (composed of ZIF and Ptzymes exhibiting different contrasts, and with a particle size of about 50–100 nm) were distributed in the brain tissue, whereas the distribution of their L-configuration counterparts was less pronounced. Similar results were confirmed by ICP-MS analysis (Fig. 5B), showing that Ptzyme@D-ZIFs showed significantly higher normalized dosage accumulation (2.20% ID g$^{-1}$) in the brain than that of Ptzyme@L-ZIFs (1.42% ID g$^{-1}$) and Ptzyme (1.12% ID g$^{-1}$). Interestingly, most of the administered Ptzyme@D-ZIFs accumulated in the kidney and liver (4.64% and 3.02% ID g$^{-1}$, respectively). Their accumulations on the kidney and liver are probably due to their hydrophilic diameter. Additionally, ICP-MS experiments were also performed to detect the concentration of Ptzyme@D-ZIFs in brain, liver and kidney at different time after administration (Supplementary Fig. 14). The results demonstrated a rapid increase in the levels of Ptzyme@D-ZIFs in the brain at 12 h, reaching a peak at the subsequent 24 h, and then gradually decreasing, returning to the initial level at 48 h. In contrast, the reduction in the liver and kidneys was not as swift, and the levels remained high even at 48 h, indicating metabolism by the liver and kidneys, a process akin to that of most other intravenous medications.

Several reports have also demonstrated the ability to tune the biological functions and pharmacokinetics of nanocomposites by introducing chirality ligands into the synthetic process[23,25,26]. Combining the results of the above in vivo and in vitro experiments, these nanozyme-integrated chiral ZIFs traverse brain microvascular endothelial cells by lysosome-independent endocytosis, indicating transcytosis as demonstrated by intracellular trafficking and BBB traverse assay. Moreover, the enhanced brain concentration of Ptzyme@D-ZIFs compared to Ptzyme@L-ZIFs may be attributed to the following reasons: (1) Ptzyme@D-ZIFs undergo endocytosis by cerebral microvascular endothelial cells and traverse the BBB through two major pathways: clathrin-mediated and caveolae-mediated endocytosis; while Ptzyme@L-ZIFs employ only clathrin-mediated endocytosis. (2) The extended residence time of Ptzyme@D-ZIFs in the bloodstream supports the accumulation of this type of nanozyme-integrated chiral ZIFs in the brain.

## In vivo therapeutic effect

To investigate the therapeutic effect of nanozyme-integrated chiral ZIFs, PD mouse models were established.

We conducted a preliminary experiment using different concentrations of Ptzyme@D-ZIF ranging from 1 to 12 mg kg$^{-1}$ day$^{-1}$, and 5 mg kg$^{-1}$ day$^{-1}$ was selected as the dose in vivo based on the therapeutic effects (Supplementary Fig. 15). As shown in Fig. 5C, the Morris water maze test was conducted to intuitively examine the behavior and memory capacity of PD mice after the administration of nanozyme formulations. As shown in Fig. 5D, PD mice exhibited random and disordered motor pathways, experiencing difficulty in locating the platform promptly. In contrast, the PD mice treated with Ptzyme@D-ZIFs reached the platform in a shorter time and a spatially oriented manner. Furthermore, compared to untreated PD mice, Ptzyme@D-ZIFs-treated PD mice showed improved target of occupancy (Fig. 5E)

and mean speed (Fig. 5F) during the treatment process, especially a significant difference in speed on day 5 (Fig. 5G), which suggested that Ptzyme@D-ZIFs dramatically rescue motor impairments and memory loss in PD mice. The therapeutic effect of Ptzyme@L-ZIFs was significantly inferior to that of Ptzyme@D-ZIFs, which was attributed to the decreased blood clearance rate and optimized distribution of Ptzyme@D-ZIFs in vivo.

Behavioral patterns were further assessed by the rotarod test and the pole test after intravenous injection of the nanozyme-integrated chiral ZIFs. As shown in Supplementary Fig. 16A, compared to healthy mice, PD mice showed a very short time (<100 s) on the rotor rod, which was attributed to neurotoxicity caused by MPTP. On the other hand, PD mice treated with Ptzyme@D-ZIFs showed a longer time on the rotating rod (>350 s), indicating that Ptzyme@D-ZIFs significantly improve the locomotor behavior of PD mice. Similar results were obtained by the pole test. As shown in Supplementary Fig. 16B, in comparison with healthy mice, PD mice exhibited a substantial increase in the duration of their running time; in contrast, Ptzyme@D-ZIFs-treated PD mice demonstrated a comparable run duration time to healthy mice in the pole test at 12 days post-injection. However, the administration of Ptzyme@L-ZIFs did not significantly decrease the running duration. In conclusion, Ptzyme@D-ZIFs alleviate MPTP-induced behavioral deficits in PD mice with higher efficiency than Ptzyme@L-ZIFs.

PD is a type of neurodegenerative disease caused by α-syn pathology, which is thought to be reflected by the degeneration of dopaminergic neurons and reduction of substantia nigra neuron density, as demonstrated by decreased TH levels (Fig. 6A)[37,38]. Therefore, we investigated whether Ptzyme@D-ZIFs could alleviate the α-syn pathology in neurons induced by MPTP. As is well known, abnormal aggregation of α-syn is a major pathological marker of PD. In particular, phosphorylated serine 129 α-syn (pS129) has been used as a biomarker to assess α-syn pathology and degree of aggregation. As expected, MPTP stimulation markedly increased pS129, while mice treated with Ptzyme@D-ZIFs showed significantly reduced α-syn pathology (Fig. 6B and Supplementary Fig. 17). Furthermore, changes in TH levels in SNpc and ST after Ptzyme@D-ZIFs administration were analyzed by immunohistochemical (Fig. 6C) and quantitative (Fig. 6D, E) analysis. The MPTP stimulation decreased TH levels, suggesting that the degeneration of dopaminergic neurons in the nigrostriatal pathway controls motor function. In contrast, PD mice treated with Ptzyme@D-ZIFs exhibited dramatically increased TH expression in both SNpc and ST. Additionally, DA content in the striatum was significantly increased after Ptzyme@D-ZIFs treatment, with levels comparable to those in healthy mice (Supplementary Fig. 18).

Furthermore, immunofluorescence analyses of pS129 and TH revealed that Ptzyme@D-ZIFs dramatically reduced pS129 levels in PD mice, while enhancing TH levels (Fig. 6F). Quantitative analysis showed a nearly 4-fold decrease in pS129/TH relative fluorescence intensity, suggesting an overall improvement in pS129-TH balance (Fig. 6G). Additionally, the TH-neurons and TH-neurons/total number of neurons were quantified both in the SNpc and ST to give a comprehensive analysis of TH levels, which suggested that Ptzyme@D-ZIFs treatment significantly reversed the loss of TH-neurons caused by MPTP stimulation (Fig. 6H-K). These data indicate that Ptzyme@D-ZIFs significantly reduce dopaminergic neuron damage and increase neuronal density in the SNpc, thus preventing pathological protein aggregation and alleviating PD symptoms.

## Therapeutic mechanisms related to oxidative injury and inflammation

Accumulation of α-syn inhibits the oxidative phosphorylation function of mitochondria, inevitably leading to overproduction of ROS and lipid peroxidation, which exacerbates oxidative damage in brain neurons. The desirable antioxidant activity of the nanozyme-

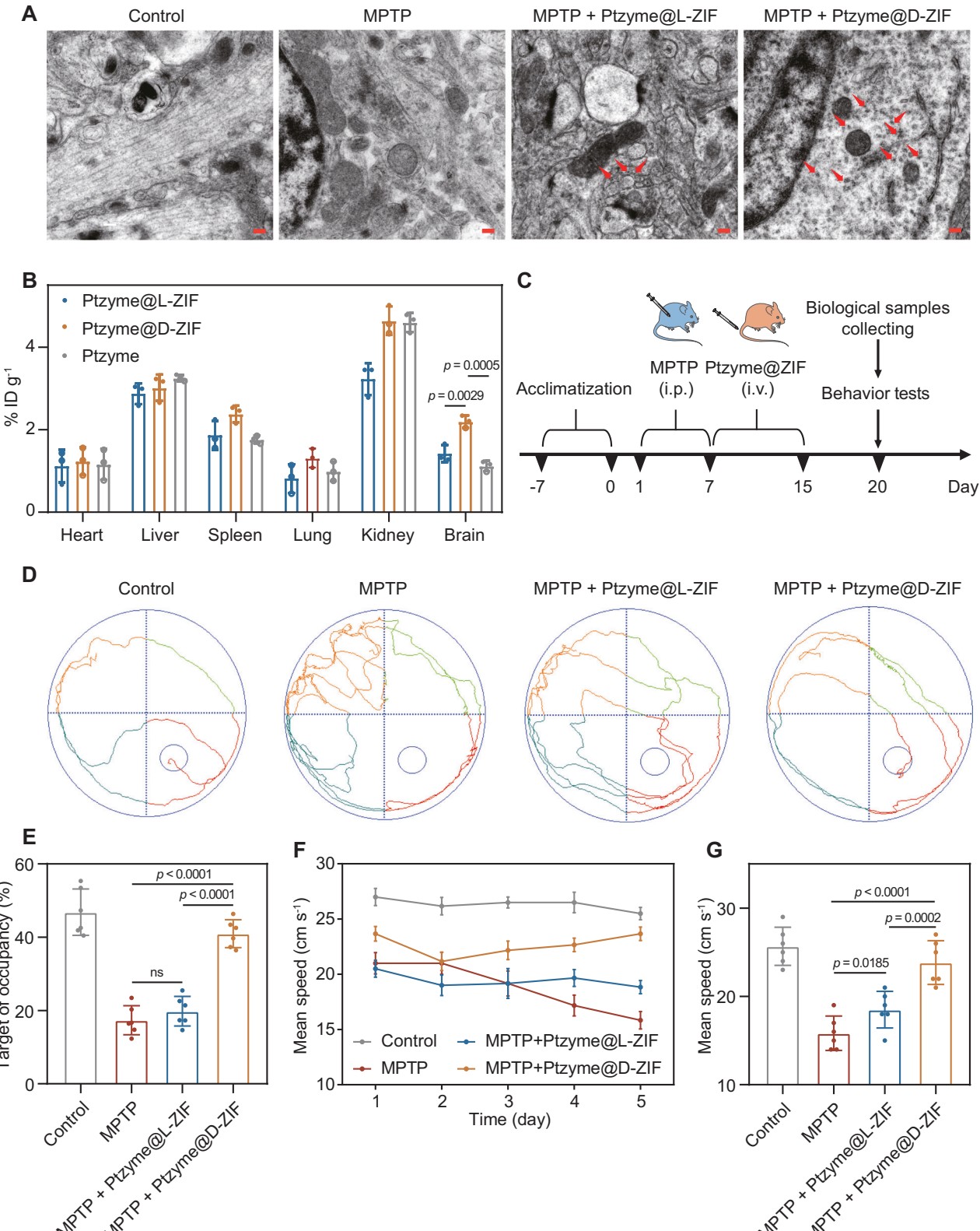

**Fig. 5 | Biodistribution and behavioral evaluation of PD mice with different treatments. A** Biodistribution of Ptzyme@D-ZIFs and Ptzyme@L-ZIFs in brain tissues at 24 h post-injection, visualized using TEM. The red arrows highlight the presence of nanoparticles in the brain tissue. Scale bars represent 0.2 μm. A representative image of three biologically independent samples from each group is shown. **B** Biodistribution of Ptzyme@D-ZIFs, Ptzyme@L-ZIFs and Ptzymes in major organs and brain detected by ICP-MS, *n* = 3 independent animals. Data represent the mean ± SD. **C** Schematic illustrating the establishment and treatment schedule of PD mice. The Morris water maze test was used to examine the behavior and memory ability of PD mice post-treatment with nanozyme-integrated chiral ZIFs. **D** The representative path tracing of mice, (**E**) the relative time spent on the target quadrant, (**F**) the mean swimming speed of mice, and (**G**) the corresponding concrete analysis at day 5, *n* = 6 independent experiments. Data represent the mean ± SD. The statistical analyses were conducted using GraphPad Prism 8.0.2. The outcomes were compared via one-way ANOVA (with Tukey's post hoc correction for multiple comparisons). "ns" indicates not significant.

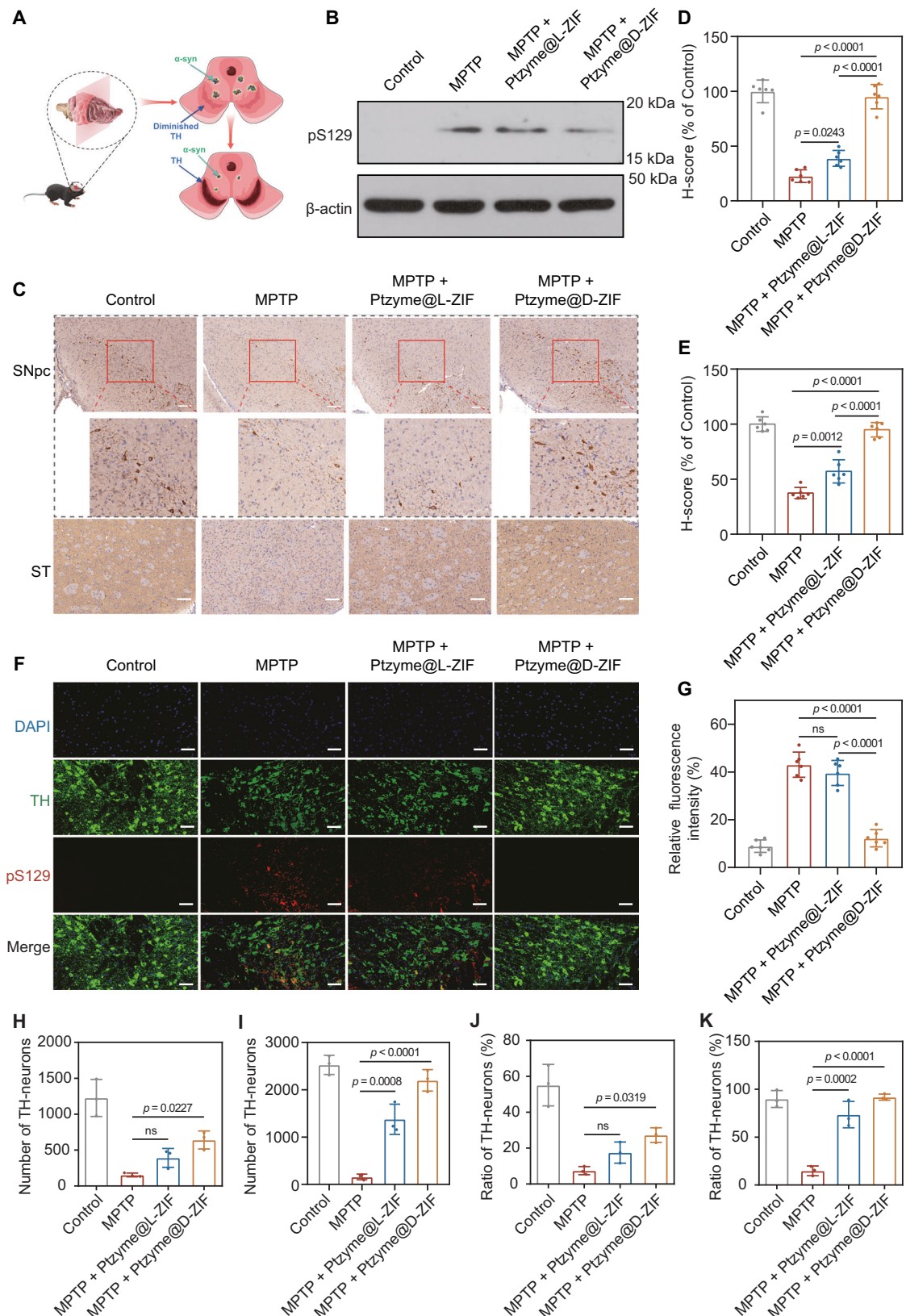

integrated chiral ZIFs in vitro inspired us to explore their neuro-protective effects in the PD mouse model based on the close rela-tionship between oxidative stress and the pathogenesis of PD. Therefore, the protective effect against oxidative damage was detected in the brains of PD mice by the levels of ROS and mal-ondialdehyde (MDA) after the administration of nanozyme-

integrated chiral ZIFs. As shown in Fig. 7A, when PD mice were treated with Ptzyme@D-ZIFs and Ptzyme@L-ZIFs, Ptzyme@D-ZIFs showed stronger ROS scavenging ability than Ptzyme@L-ZIFs. Phospholipids are essential for cell membrane integrity but are vulnerable to excess ROS attacks. To assess lipid peroxidation injury, we analyzed the levels of MDA in the brains of PD mice

**Fig. 6 | Pathological evaluations of the brains of PD mice post-treatment with nanozyme-integrated chiral ZIFs. A** Schematic illustration of the pathological changes of PD with respect to the accumulation of α-syn and diminished TH after treatment with nanozyme-integrated chiral ZIFs. **B** Western blotting showing the expression changes of pS129 in the brains of PD mice following treatment with nanozyme-integrated chiral ZIFs. A representative blot of three independent experiments is shown. Changes of TH levels detected by (**C**) immunohistochemical and the corresponding quantitative analysis in (**D**) SNpc and (**E**) ST, $n = 6$ independent samples in independent animals. Data represent the mean ± SD. The scale bars are 100 μm. A representative image of six biologically independent samples from each group is shown. **F** Co-immunoreactivity analysis of the brain sections

from PD mice by immunofluorescence, stained with anti-pS129 antibody (red) and anti-TH antibody (green) and (**G**) the corresponding quantitative results expressed as the relative fluorescence intensity of pS129 and TH, $n = 6$ independent samples in independent animals. A representative image of six biologically independent samples from each group is shown. Data represent the mean ± SD. The scale bars are 50 μm. Quantification of TH-neurons in the SNpc (**H**) and ST (**I**), and the number ratio of TH-neurons and total neurons in the SNpc (**J**) and ST (**K**) $n = 3$ independent samples in independent animals. Data represent the mean ± SD. The statistical analyses were conducted using GraphPad Prism 8.0.2. The outcomes were compared via one-way ANOVA (with Tukey's post hoc correction for multiple comparisons). "ns" indicates not significant.

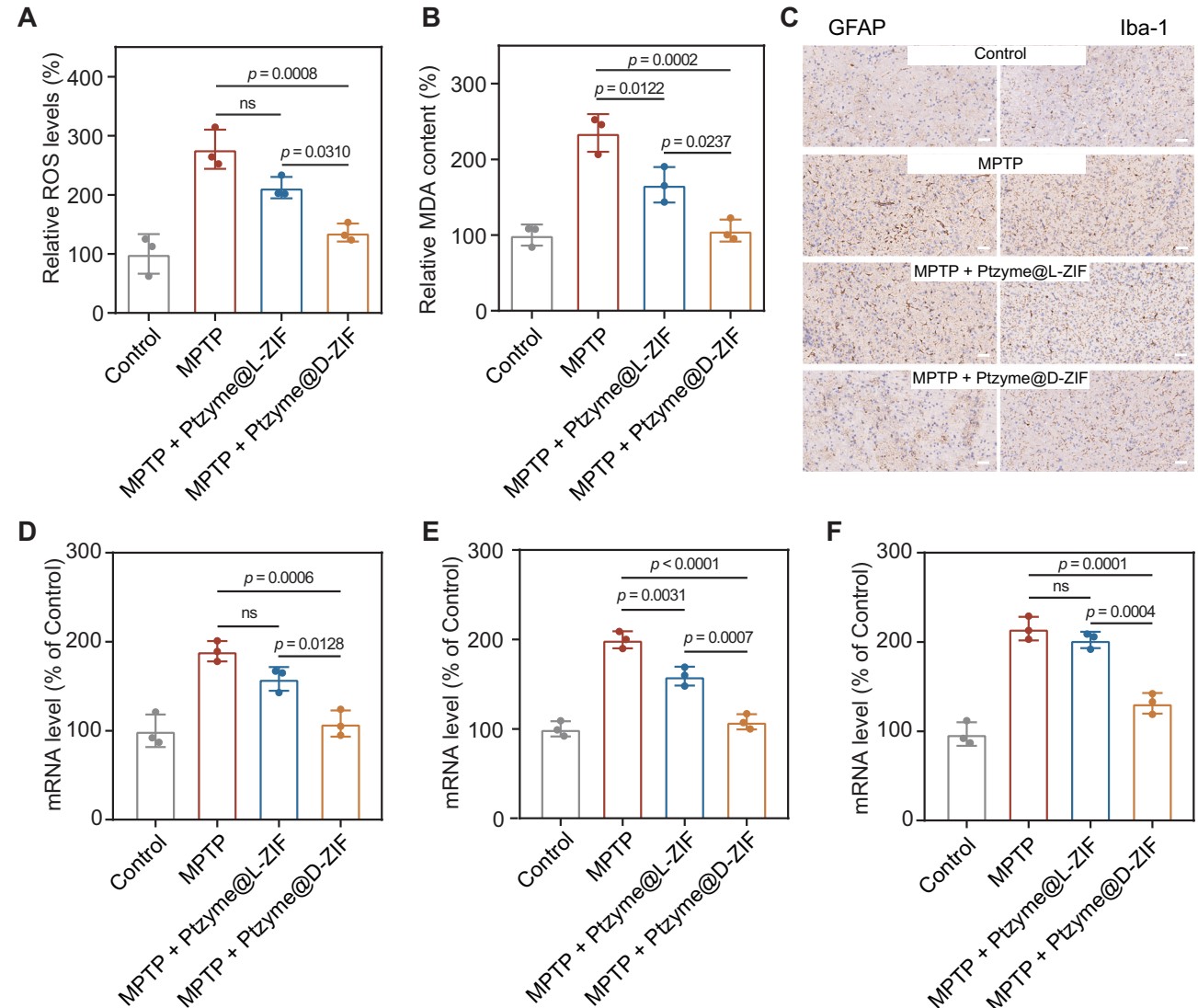

**Fig. 7 | Therapeutic mechanisms of nanozyme-integrated chiral ZIFs in PD mice with respect to oxidative damage and inflammation. A** Levels of reactive oxygen species (ROS) and (**B**) malondialdehyde (MDA) in the brains of PD mice after treatment with nanozyme-integrated chiral ZIFs, $n = 3$ independent samples in independent animals. Data represent the mean ± SD. **C** GFAP and Iba-1 expressions in brain sections of PD mice from various groups were analyzed by immunohistochemical staining. The scale bars are 50 μm. A representative image of three

biologically independent samples from each group is shown. Inflammatory factors of (**D**) *TNF-α*, (**E**) *IL-6* and (**F**) *IL-1β* in the brain tissues of different groups, $n = 3$ independent samples in independent animals. Data represent the mean ± SD. The statistical analyses were conducted using the GraphPad Prism 8.0.2. The outcomes were compared via one-way ANOVA (with Tukey's post hoc correction for multiple comparisons). "ns" indicates not significant.

(Fig. 7B). A significant decrease in MDA content was achieved after Ptzyme@D-ZIFs treatment compared to MPTP-induced PD mice, which was consistent with the detection of ROS levels. These results indicate that Ptzyme@D-ZIFs reduce oxidative damage and avoid dopaminergic neuron degeneration by scavenging ROS. In addition, Ptzyme@D-ZIFs exhibit stronger ROS scavenging ability than Ptzyme@L-ZIFs due to their longer half-life in vivo and superior accumulation in the brain.

Excessive ROS in the brain of PD mice, along with the expression of inflammatory factors, induces an inflammatory response, leading to

the proliferation and activation of a wide range of glial cells. Among these glial cells, astrocytes and microglia are essential inflammatory cells, and their activation signals inflammatory responses in the brain. Since glial fibrillary acidic protein (GFAP) and Iba-1 are makers of astrocytes and microglia, respectively, their expression levels were measured by immunohistochemical staining. As shown in Fig. 7C, the expression levels of GFAP and Iba-1 were upregulated in MPTP-induced PD mice. In contrast, their expression levels were significantly decreased after Ptzyme@D-ZIFs administration. This alteration in expression level was also evidenced by the semiquantitative analysis of GFAP- and Iba-1-positive regions (Supplementary Fig. 19), indicating a decrease in the number of astrocytes and microglia after Ptzyme@D-ZIFs treatment. Additionally, tumor necrosis factor-α (TNF-α) and the relevant interleukins (ILs) such as IL-6 and IL-1β were also determined to assess the inflammatory levels in lesions, suggesting that Ptzyme@D-ZIFs significantly reduce the gene expression levels of *TNF-α* (Fig. 7D), *IL-6* (Fig. 7E) and *IL-1β* (Fig. 7F), which were enhanced under the stimulation of MPTP. It is noteworthy that secretion of these factors was reduced to levels similar to those in the control group. These results indicate that Ptzyme@D-ZIFs show a remarkable ability to mitigate the oxidative damage of brain lesions, thus reducing the inflammatory response and expression of inflammatory factors in PD model mice, benefiting from their brain accumulation and ROS removal capacity.

## Transcriptomics analyses

The above studies demonstrated an obvious therapeutic mechanism with respect to ROS and inflammation at the histopathological level. Transcriptomic analyses were conducted to comprehensively understand the therapeutic process of Ptzyme@D-ZIFs and elucidate the underlying therapeutic mechanisms at the gene level. Unguided analysis of differential expression was illustrated by a Venn diagram (Fig. 8A) and a volcano plot (Fig. 8B). In the Venn diagram, 12,487 genes were co-expressed in the MPTP and MPTP + Ptzyme@D-ZIF treated groups, while 208 genes were exclusive in the MPTP + Ptzyme@D-ZIF treated group. In addition, the volcano plots suggested 365 dramatically differentially expressed genes (DEGs), of which 181 were downregulated and 184 were up-regulated. Gene Ontology (GO) enrichment analysis (Fig. 8C) showed significant differences in animal behavior and regulation of location, consistent with behavioral assessment. In addition, signaling pathways with respect to neurogenesis, neuron differentiation and the biological processes such as regulation of biological quality, trans-synaptic signaling, and nervous system development (Supplementary Table 1) were also considerably altered after treatment, which is closely related to the development of PD. Moreover, the Kyoto Encyclopedia of Genes and Genomes (KEGG) enrichment analysis (Fig. 8D) suggested a remarkable change in the disease pathway of Parkinson's. Details in the pathway were investigated through KO analysis (Supplementary Fig. 20), in which the expression levels of *Parkin*, *SNCA*, *PINK1*, *DJ-1*, and *LRRK2* were adjusted after Ptzyme@D-ZIFs treatment. These genes have been shown in the literature to play essential roles in the proteolytic process via the ubiquitin-proteasome system (UPS). Therefore, adjusting these genes to normal levels will prohibit protein homeostasis imbalances and inhibit abnormal aggregation of α-syn. This was also demonstrated by biochemical analysis of α-syn as described above. KEGG enrichment analysis suggested that the NF-κB signaling pathway and the PI3K-Akt signaling pathway are strongly associated with the therapeutic mechanism of Ptzyme@D-ZIFs (Fig. 8D). The activation of the NF-κB signaling pathway by ROS and its resulting induction of a systemic inflammatory response are widely acknowledged. Moreover, the PI3K-Akt signaling pathway has been reported to be highly associated with regulating cell proliferation and apoptosis. More importantly, molecular functional ontology analysis showed that ion transmembrane transport and ion channel activity were significantly activated after

Ptzyme@D-ZIFs treatment (Supplementary Table 2). Excessive ion deposition in the brain is considered a hallmark of PD patients, and thus, the activation of ion channels may signal freedom from PD symptoms[39]. In addition, the activation of ion channels suggested a relationship between the treatment mechanism and the signaling pathway of ferroptosis. Furthermore, heatmap analysis revealed that the expression levels of several key genes that promote ferroptosis, such as *FDFT1*, *Nox1*, and *CS*, were significantly decreased after Ptzyme@D-ZIFs treatment (Fig. 8E). In contrast, ferroptosis suppressor genes such as *TTC35* and *GPX4* were increased (Fig. 8F), indicating that the ferroptosis program is closely related to the therapeutic mechanisms of Ptzyme@D-ZIFs. These results suggest that Ptzyme@D-ZIFs may exert its therapeutic effects by removing ROS, relieving inflammation, and inhibiting neuronal apoptosis and ferroptosis induced by MPTP.

## Verification of therapeutic mechanism and biosafety analysis

To verify the efficacy of Ptzyme@D-ZIFs in ameliorating neuronal apoptosis and ferroptosis, in vitro studies were conducted using SH-SY5Y cells, which are widely used to construct PD phenotype under 1-methyl-4-phenylpyridinium (MPP$^+$) stimulation. The cytotoxicity of Ptzyme@D-ZIFs was assessed by MTT assay (Supplementary Fig. 21A). The nanozyme-integrated chiral ZIFs showed negligible toxicity to SH-SY5Y cells at doses lower than 80 μg mL$^{-1}$, and this dose was selected as the given dose in subsequent experiments. Furthermore, a neuroprotective effect was detected at this dose. As shown in Supplementary Fig. 21B, Ptzyme@D-ZIFs showed the desired protective effect against MPP$^+$-induced cell death, restoring cell viability to more than 85% of the control group.

Previous studies have shown that the neurotoxin MPP$^+$ specifically targets dopaminergic neurons, generating excessive ROS, inhibiting the normal oxidative phosphorylation function of mitochondria and causing cell death. Therefore, we detected intracellular ROS levels using DCFH-DA, a widely used ROS-sensitive fluorescent dye. As shown in Fig. 9A, Ptzyme@D-ZIFs significantly reduced DCFH-DA fluorescence intensity in SH-SY5Y cells, indicating that Ptzyme@D-ZIFs effectively remove redundant ROS induced by MPP$^+$. Further quantitative analysis showed that DCFH-DA intensity increased by over 2-fold after MPP$^+$ stimulation but decreased to levels similar to the control group after treatment with Ptzyme@D-ZIFs (Supplementary Fig. 21C). Moreover, the anti-apoptotic effect of Ptzyme@D-ZIFs in SHSY-SY cells was verified by flow cytometry and caspase 3/7 detection: the apoptosis (including early and late apoptosis) ratio was increased by 81.42% after MPP$^+$ stimulation, while after Ptzyme@D-ZIFs treatment, the apoptosis ratio decreased to 8.02%, similar to that of control cells (3.10%) (Fig. 9B and Supplementary Fig. 22). Similar results were seen in the caspase 3/7 assay, with a dramatic reduction in MPP$^+$-induced apoptosis in cells protected by Ptzyme@D-ZIFs (Supplementary Fig. 21D).

Moreover, transcriptome analysis revealed alterations in the expression levels of several important genes related to the pathway of ferroptosis, prompting a more in-depth investigation into the therapeutic mechanism related to ferroptosis. As shown in Fig. 9C, GPX4 expression was markedly decreased after MPP$^+$ stimulation but was markedly improved by the addition of Ptzyme@D-ZIFs due to their ROS scavenging ability. Furthermore, the level of MDA was dramatically reduced in Ptzyme@D-ZIF-treated cells compared to MPP$^+$-stimulated cells (Fig. 9D). Downregulation of GPX4 and elevated ROS levels usually leads to accumulation of MDA, an important indicator of ferroptosis. Therefore, these results indicate that Ptzyme@D-ZIFs prevent the development of ferroptosis by regulating GPX4 expression to normal levels and suppressing the overproduction of MDA.

Finally, fluctuations in mitochondrial membrane potential are a common marker for both ferroptosis and apoptosis. When cells were in a normal state, the JC-1 probe was present as aggregates in the

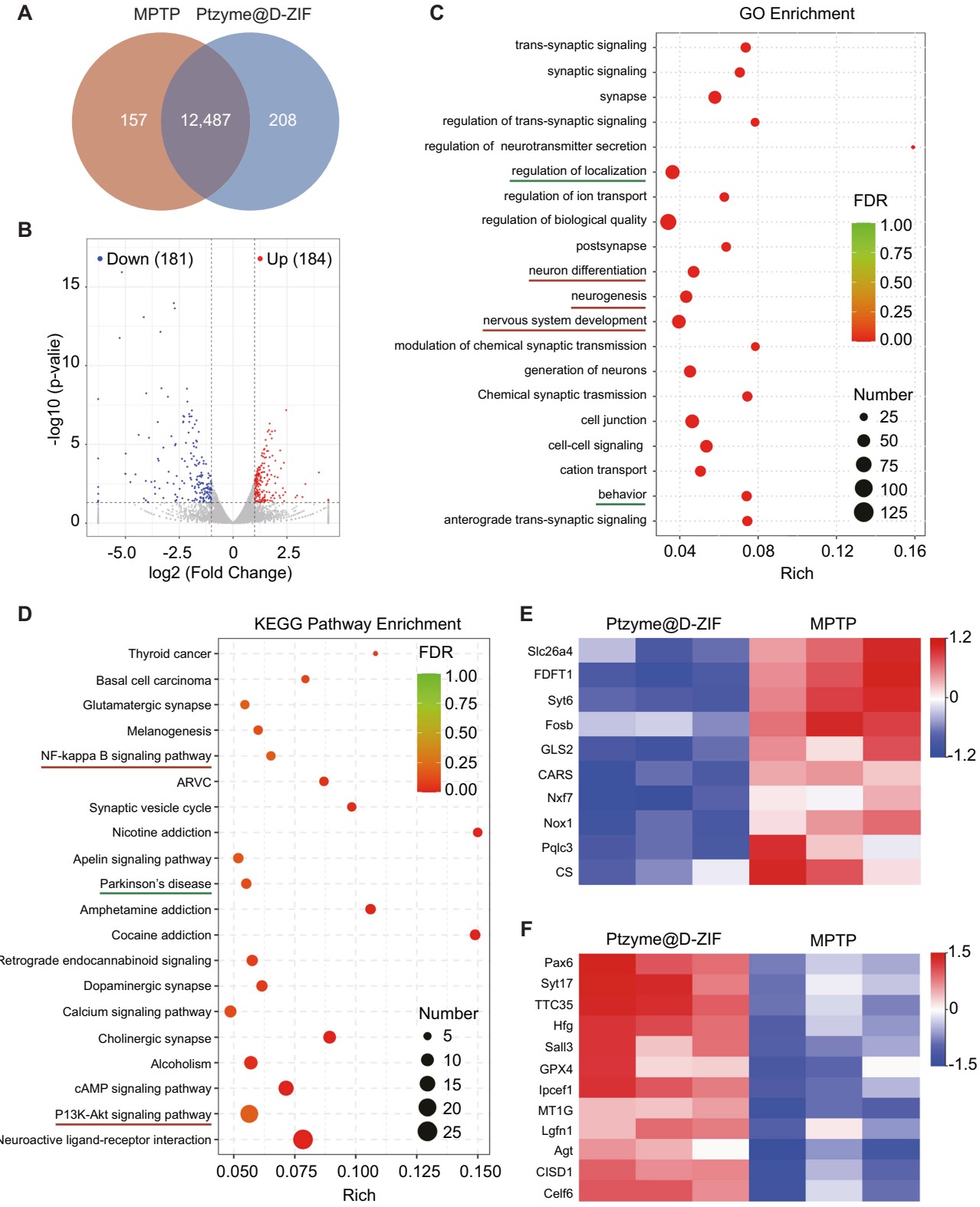

**Fig. 8 | Transcriptomics analyses of PD mice with or without Ptzyme@D-ZIF therapy. A** Venn diagram of transcriptomic profiles on PD mice with or without Ptzyme@D-ZIF treatment. **B** Volcano plots illustrate the dramatically differentially expressed genes (DEGs) after treatment with Ptzyme@D-ZIF. (**C**) GO enrichment and (**D**) KEGG pathway enrichment analysis of the identified DEGs, which show the 20 most remarkably enriched pathways. The rectangular box circled the enriched target signaling pathways. Heat maps of dramatically (**E**) downregulated and (**F**) upregulated genes for the analysis of ferroptosis.

mitochondrial matrix and emitted red fluorescence. During apoptosis and ferroptosis, the mitochondrial membrane potential decreased and green fluorescence was observed. Strong green fluorescence was detected after MPP⁺ stimulation, suggesting that the mitochondria-dependent apoptosis and ferroptosis mechanisms were activated. On the other hand, upon treatment with Ptzyme@D-ZIFs, strong red fluorescence was detected, suggesting a protective effect against mitochondrial dysfunction in cells (Fig. 9E and Supplementary

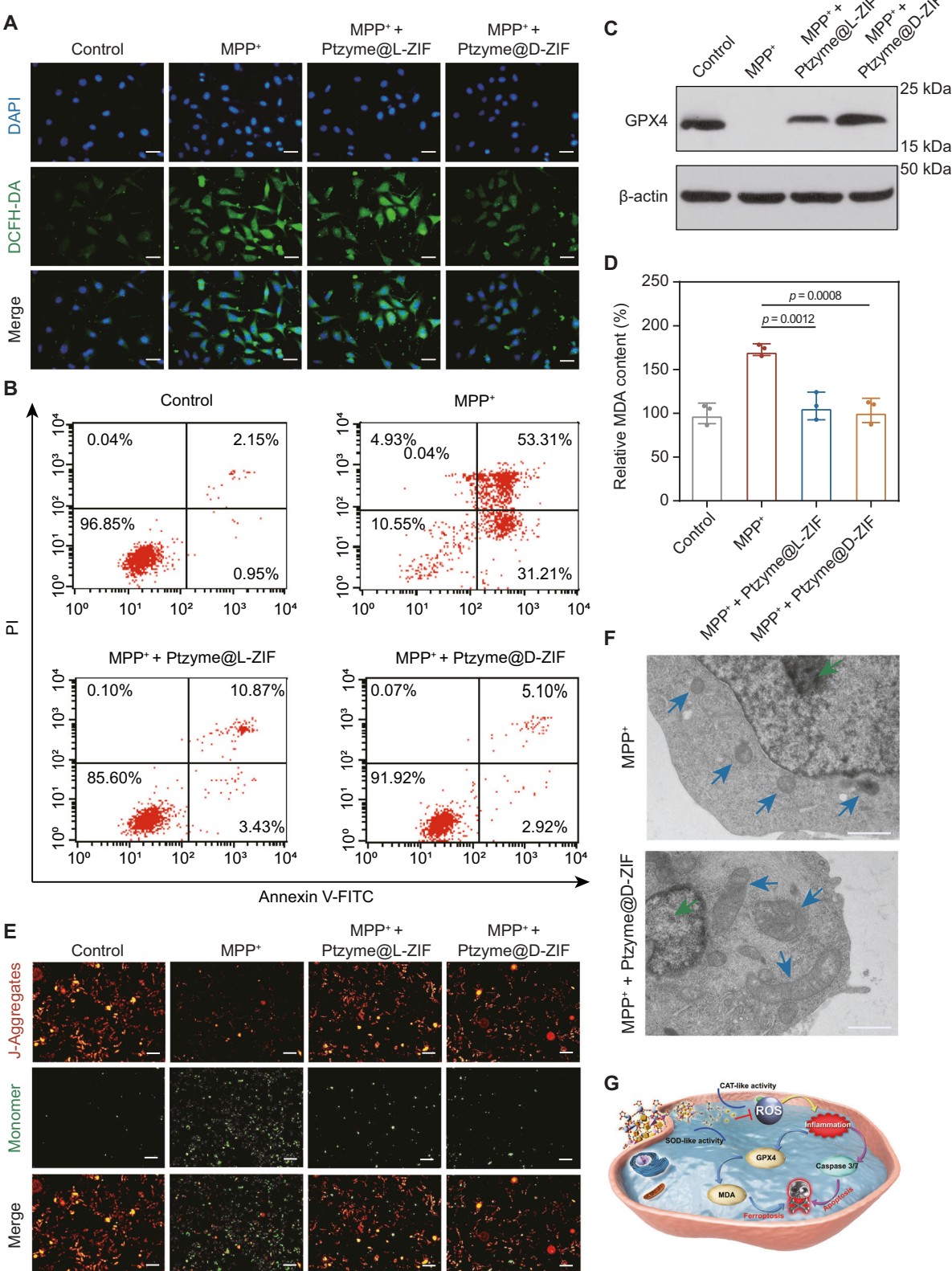

Fig. 21E). Similarly, TEM images of MPP⁺-stimulated cells showed clear features of apoptosis and ferroptosis, such as chromatin condensation and mitochondrial contraction, which were significantly alleviated by Ptzyme@D-ZIFs treatment (Fig. 9F). The above results further confirmed the conclusion of the transcriptome analysis at the cellular level, namely that Ptzyme@D-ZIFs effectively alleviate the anabatic inflammatory response by trapping intracellular ROS and thus mitigate cellular damage via both apoptosis and ferroptosis pathways (Fig. 9G).

In addition, the biosafety of nanomedicines is one of the main factors limiting their application. Therefore, we conducted a systematic evaluation of the impacts of nanozymes integrated chiral ZIFs on vital organs (heart, liver, spleen, lung, and kidney) in healthy mice,

**Fig. 9 | Validation of the therapeutic mechanism of nanozyme-integrated chiral ZIFs on MPP+ induced PD cell models in SH-SY5Y cells. A** Intracellular levels of ROS induced by MPP+ were detected with the ROS fluorescent probe DCFH-DA in SH-SY5Y cells following treatment with or without nanozyme-integrated chiral ZIFs. The scale bars are 20 μm. A representative image of three biologically independent samples from each group is shown. **B** The analysis of cell apoptosis was performed using Annexin V-FITC/PI staining and flow cytometric detection in untreated and nanozyme-integrated chiral ZIFs treated SH-SY5Y cells incubated with 2 mM MPP+. **C** GPX4 expression levels through Western blotting and (**D**) analysis of MDA content using an assay kit in untreated and nanozyme-integrated chiral ZIFs treated SH-SY5Y cells incubated with 2 mM MPP+, $n = 3$ independent experiments. Data represent the mean ± SD. A representative blot of three independent experiments is shown. **E** Analysis of mitochondrial membrane potential in untreated and

nanozyme-integrated chiral ZIFs treated SH-SY5Y cells incubated with 2 mM MPP+. The scale bars are 100 μm. A representative image of three biologically independent samples from each group is shown. **F** Morphology analyses were conducted using TEM on untreated and Ptzyme@D-ZIFs treated SH-SY5Y cells incubated with 2 mM MPP+. The green and blue arrows indicate the positions of the nucleus and mitochondria, respectively. The scale bars are 1 μm. A representative image of three biologically independent samples from each group is shown. **G** Schematic illustrating that Ptzyme@D-ZIFs relieve anabatic inflammatory response by scavenging intracellular ROS, thus mitigating cellular damage through both apoptosis and ferroptosis pathways. The statistical analyses were conducted using GraphPad Prism 8.0.2. The outcomes were compared via one-way ANOVA (with Tukey's post hoc correction for multiple comparisons).

along with an assessment of biochemical blood indices in PD mice. As shown in the hematoxylin and eosin staining result, sections of major organs after the nanozyme-integrated chiral ZIFs administration presented no significant inflammation or other pathological features (Supplementary Fig. 23A). Furthermore, compared to untreated PD mice, the expression of ALT (Supplementary Fig. 2B), AST (Supplementary Fig. 23C), BUN (Supplementary Fig. 23D), and CRE (Supplementary Fig. 23E) in the serum of mice was not significantly altered after the administration of nanozyme-integrated chiral ZIFs, indicating the negligible influence on the functions of liver and kidney. These results suggest the biosafety potential of Ptzyme@D-ZIFs for future biomedical applications.

After fully investigating the mechanisms of in vivo action and biosafety of our nanozyme-integrated chiral ZIFs on PD biological models, we believe that a brief research review and future prospects in related fields are necessary. There has been a long history of research into mitigating the symptoms of PD by delivering antioxidants to the brain because of the close association between ROS and subsequent neuroinflammation in the occurrence and development of PD. While natural enzymes were initially used for their antioxidant properties to reduce neuroinflammation, their inefficacy in crossing BBB and in vivo instability were notable drawbacks for further clinical applications[40]. The discovery of nanozymes has spurred investigations into the brain delivery of these agents to alleviate PD symptoms. For example, Singh et al. demonstrated that $Mn_3O_4$ nanozyme effectively and consistently scavenge ROS in SH-SY5Y cells, offering a potential treatment for PD cell models [41]. In recent years, continuous studies have been conducted to explore the potential mechanisms of PD therapy based on the antioxidant activities of nanozymes, enhancing their therapeutic potential in regulating PD disease progressions, including as α-synuclein transmission, inflammasome assembly, and the prevention of pyroptosis.

This study is an in-depth exploration of the mechanisms in the treatment of PD using nanozyme-intergrated chiral ZIFs Previous studies primarily focus on the enzyme-like function of nanozymes in ROS elimination and PD neuropathology improvement. Our study further investigated the influence of nanozyme structure and validated their therapeutic application potential in PD by exploring the association between the chiral properties of nanozyme-integrated ZIFs and the traversing modes in BBB crossing, the action mechanism in vivo, and biosafety evaluations. Prior study has reported the impact of the chirality of ligands in nanoparticles on the pharmacokinetics and metabolism of drugs, as well as the cellular responses upon exogenous stimulations[23]. Herein, based on the above studies, we further explain the discrepancies in cytology and in vivo mechanisms of action of nanozyme-intergrated chiral ZIFs. We are currently conducting subsequent studies and anticipate other researchers to broaden the scope and depth of our research. Further investigations are worthy of addressing distinct pathways utilized by various chiral nanozymes for BBB penetration. A critical aspect of this inquiry is identifying the cellular receptors that may be involved in facilitating BBB permeation.

Additionally, the exploration of whether alternative chiral ligands elicit similar biological responses and the capability to pinpoint optimal ligands and chiral configurations through sophisticated techniques, such as metadata analysis and material genomics, are also imperative for advancing future studies in this field.

In summary, we have successfully developed two types of nanozyme-integrated chiral ZIFs by embedding Ptzymes in L- and D-chiral ZIF shells. These results demonstrate that the Ptzyme@D-ZIFs not only exhibit stronger cascade SOD- and CAT-mimetic activity, but also present effective BBB traverse ability, better brain accumulation and plasma stability, showing more satisfactory potential in eliminating ROS and inflammation in the brain. Furthermore, in vitro and in vivo studies demonstrated that the therapeutic effects of Ptzyme@D-ZIFs are achieved by inhibiting apoptosis and ferroptosis pathways on injured neurons. Moreover, these nanozyme-integrated chiral ZIFs show negligible biological toxicity during the therapeutic process. In conclusion, these data provide proof-of-concept that nanozyme-based chiral nanomaterials are successfully developed against PD through differential metabolism and multi-mechanism therapy. In the future, we will further describe the metabolic differences, biological activities, and in vivo mechanisms of action of chiral nanomaterials. We believe this research will help design more chiral nanomaterials for pharmaceutical applications[42,43].

## Methods

### Synthesis of Ptzyme@D-ZIF and Ptzyme@L-ZIF

2-methylimidazole (785 mM), 5 mM D-histidine and 1 mg Ptzyme were dissolved in 2.0 mL deionized water containing 0.025 wt% CTAB and stirred at 500 r/min for 5 min, then 0.5 mL $Zn(NO_3)_2 \cdot 6H_2O$ aqueous solution (97.5 mM) was added. After stirring for 5 min, the mixture was left undisturbed for 3 h at room temperature. Finally, the prepared Ptzyme@D-ZIF was collected by centrifugation. After that, Ptzyme@D-ZIF was dispersed into the corresponding D-histidine solution (1 mL, 10 mM) and ultrasonicated for 3 min, the final Ptzyme@D-ZIF was collected by centrifugation at 3500 g for 10 min. Ptzyme@L-ZIF was synthesized as same as Ptzyme@D-ZIF, except L-histidine was instead of D-histidine.

### BBB permeability

To prepare FITC-Ptzyme@D-ZIF and FITC-Ptzyme@L-ZIF, 100 mg of BSA and 20 mg of FITC were dissolved in 50 mL of PBS (40 mM, pH 8.0). The mixture was stirred in the dark at room temperature for 48 h. Then, the mixed solution was dialyzed against distilled water using an Amicon® Ultra-15 ultrafiltration tube (Millipore, MA) until no absorbance at 488 nm was detectable in the supernatant. Finally, the solution was lyophilized to obtain FITC-BSA. The subsequent steps were similar to the procedure used for Ptzyme@ZIF preparation. The BBB model in vitro was constructed using BEnd.3 cells and transwell chambers. Briefly, co-culturing of SH-SY5Y cells in the lower chamber and bEnd.3 cells in the upper chamber was carried out for 2 days until

the formation of a tight junction by bEnd.3 cells. Subsequently, the culture media was supplemented with FITC-Ptzyme@D-ZIFs and FITC-Ptzyme@L-ZIFs (40 µg mL$^{-1}$) to evaluate the BBB permeability. After incubation in the dark for 12 h, the cells in the lower chamber (SH-SY5Y cells) were collected and the fluorescence intensity was measured at 485 nm using an Infinite F200 Pro microplate reader (TECAN, Switzerland).

## Intracellular localization

Briefly, bEnd.3 cells were inoculated in 6-well plates with a density of $1.8 \times 10^5$ cells/well and cultured in a cell incubator overnight. The cells were stained by Hoechst 33342 (blue) and Lyso Tracker (red), and then incubated with FBS-free DMEM containing FITC-Ptzyme@D-ZIFs or FITC-Ptzyme@L-ZIFs (40 µg mL$^{-1}$, green) for 2 and 4 h, respectively. Finally, the coverslip was used for the detection on an LSM 880 confocal laser scanning microscope (Carl Zeiss Microscopy LLC, Jena, Germany) to analyze the real-time localization of these nanozyme-integrated chiral MOF platforms in bEnd.3 cells.

## Intracellular internalization mechanisms

Briefly, bEnd.3 cells were inoculated in 96-well plates at an initial density of $8 \times 10^3$ cells/well and cultured at 37 °C. Afterwards, the cells were cultured with FITC-Ptzyme@D-ZIFs or FITC-Ptzyme@L-ZIFs (40 µg mL$^{-1}$) for 4 h in the corresponding culture conditions. After medium removal, the cells were washed with PBS five times and subjected to fluorescence intensity detection via a microplate reader.

## Animals

All of the animal experiments were approved by the Institution Animal Ethics Committee of Zhengzhou University (license No. ZZU-LAC20210625[07]). C57BL/6 male mice of 5-week-old (SiPeifu Biotechnology Co., Ltd., Beijing, China) were fed in cages with controlled temperature and humidity. We opted for male animals due to their heightened sensitivity to MPTP, facilitating the construction of Parkinson's disease (PD) models with greater ease and reliability. All mice kept free access to water and food. Before subsequent experiments started, all mice were left to adapt for 7 days in the animal house with 12 h light/dark cycles to simulate the natural environment of day and night.

## Blood clearance rate

The PD-like phenotype mice were established as discussed above. Afterwards, the mice were intravenously injected with Ptzyme@D-ZIFs and Ptzyme@L-ZIFs (5 mg/kg, $n = 6$), respectively. Blood was collected intravenously at 10 min, 0.5, 1, 2, 4, 8, 24, 48, and 72 h after injection. The blood was collected by an anticoagulant tube with heparin sodium. Immediately after blood collection, the blood was centrifuged (600 g, 10 min) to obtain plasma and transferred to a new EP tube. Finally, the SOD units in plasma were detected by WST Total Superoxide Dismutase Assay Kit (S0101, Beyotime) and the half-lives of Ptzyme@D-ZIF and Ptzyme@L-ZIF were calculated.

## Morris water maze test

The maze (depth: 70 cm, width: 150 cm) is composed of four quadrants, a platform, and a circular pool with black opaque plastic. The temperature of water in the circular pool was kept at 24 - 26 °C. On the first 7 days, the mice ($n = 6$) were placed at the four quadrants respectively and trained to reach the platform independently. The mouse failed to find the platform was guided to and remained there for 1 min. After the training process, the mice were performed a spatial memory test. Briefly, each mouse was placed into the water opposite the platform to swim freely for 4 min and the fixed platform used in the training experiment was removed from the maze. The spatial acuity was expressed as the swimming speed and relative time the mouse spent on the targeted quadrant (target of occupancy %).

## Rotarod test

In the rotarod test, mice were pretrained on the fatigue rotating rod (ZB-200, Cheng Du Technology & Market Co., China) at a rotation speed of 8, 16, 24 rpm and constant acceleration from 4 to 44 rpm for continuous 5 min, respectively. Following MPTP induction and different treatments, mice were tested on the rotating rod at a speed of 40 rpm. The latency for each mouse on the rod was recorded and the final score was considered as the average time of three tests.

## Pole test

Pole test was performed on the first day of Ptzyme@D-ZIFs injection and recorded every other day for 7 times. The duration time of each mouse climbing from the top of a 50 cm-length pole to the end was tested and recorded at least three times.

## Histological analysis

The brains of PD mice were fixed with 4% formaldehyde for 72 h, after which the SNpc and ST tissues were obtained and embedded in paraffin. Subsequently, the issues were sectioned into 10 µm with ultra-thin slices (Leica UC7, Germany) and then stained for the subsequent histological analysis. The sections were placed in the EDTA (pH = 9.0) solution for antigen retrieval, with medium heat for 8 min to boiling, a ceasefire for 8 min to keep warm, and then a medium heat for another 7 min. After blocking endogenous peroxidase with 3% H$_2$O$_2$ for 25 min and blocking with 3% BSA for 40 min at room temperature, antibodies against TH (GB12181, Servicebio, China, dilution 1:500), pS129 (ab51253, Abcam, China, clone EP1536Y, dilution 1:500), GFAP (GB12096, Servicebio, China, dilution 1:1000), Iba-1 (GB13105-1, Servicebio, China, dilution 1:1000) were incubated with the sections at 4 °C for 24 h. and then HRP-labeled secondary antibodies for 40 min at room temperature. Then, the sections were incubated with DAB reagent and subjected to hematoxylin counterstaining and photographed via the microscope.

For immunohistochemical analysis, the positive area and staining intensity were semi-quantified via the digital tissue section scanner (Aperio VERSA8, Leica) and image analysis system (Servicebio, China), and expressed as the percentage of H-score of other groups to the control group. H-score = (percentage of weak intensity area × 1) + (percentage of moderate intensity area × 2) + (percentage of strong intensity area × 3). For immunofluorescence analysis, the average fluorescence intensity of the target area of each section was quantified by Indica Labs-Area Quantification FLv 2.1.2 module in Halo v3.0.311.314 analysis software (Indica labs, U.S.A). Additionally, the stereological quantification of TH-neurons in the SNpc and ST region of PD mice were performed using Aipathwell software (Servicebio, China).

## Immunoblot analysis

The concentrations of protein samples obtained from SNpc tissues or SH-SY5Y cells were detected using BCA assay (G2026, Servicebio), after which the samples were subjected to SDS-polyacrylamide gels (12%) for separation and PVDF membrane for transfer. Then, the membranes were blocked using 5% BSA in PBST at room temperature. Afterwards, the samples were incubated with primary antibody (GPX4, ab125066, Abcam, clone EPNCIR144, dilution 1:1000) for 24 h, and then secondary antibody labeled with HRP (GB23303, Servicebio, dilution 1:3000). Subsequently, the target antigens were imaged by chemiluminescence imaging system (2500, Tanon) and Alpha software processing system (alphaEaseFC, Alpha Innotech) was applied for analyzing the optical density of the bands.

## Statistical analysis

All statistical analyses were conducted using the GraphPad Prism 8.0.2 in a blinded manner. The differences in means between different groups were compared via Student's $t$-test, one-way ANOVA (with Tukey's post hoc correction for multiple comparisons), or two-way

ANOVA test (with Bonferroni's multiple comparisons test). The specific statistical method and statistical analysis results for each experiment were listed in the corresponding figure legends.

## Reporting summary

Further information on research design is available in the Nature Portfolio Reporting Summary linked to this article.

## Data availability

All data needed to evaluate the conclusions in the paper are present in the paper and/or the Supplementary Information and Source Data. The accession code of National Center for Biotechnology Information concerning the raw sequence reads in RNA-seq data is PRJNA1032155. Additional data is available from the corresponding authors upon request. Source data are provided with this paper.

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

## Acknowledgements

We are grateful for the financial support by the National Natural Science Foundation of China (Grant No. 82122037 K.F., 81930050 X. Y., 22121003 X. Y.), National Key Research and Development Program of China (No. 2021YFC2102900, K.F.), CAS Project for Young Scientists in Basic Research (YSBR-089, K.F.), CAS Interdisciplinary Innovation Team (JCTD-2020-08, K.F.), Henan Province Key Research and Promotion Project (222102310567, W.J.), and the Youth Science Foundation of Henan Province (232300421285, Q.L.). We thank Prof. Xianghui Xu from Hunan University for his beneficial discussion on manuscript.

## Author contributions

K.F. X.Y. and Q.L. designed, conceived, and supervised the project and wrote the manuscript. W.J. Q.L. designed and performed most of the experiments, analyzed data, and contributed to manuscript preparation. R.Z. and J.R.L. assisted in the in vivo experiments. W.J. Q.Y.L. J.Y.L. and X.Z. conducted the mechanism analysis of Ptzyme. All authors critically revised the manuscript and approved the submitted version.

## Competing interests

The authors declare no competing interests.
