## [Peer Review File · Nature Communications]

Reviewers' Comments:

Reviewer #1:

Remarks to the Author:

In this manuscript, the Authors report on ultra-small platinum nanozymes (Ptzymes) incorporated chiral zeolite imidazolate frameworks that can potentially be used as inhibitors of ferroptosis/apoptosis for the treatment of Parkinson's disease. In general, the experimental design is of scientific rigour; however data interpretation and presentation should be improved. The importance of finding new reagents to treat Parkinson's disease and the level of novelty of this work would be sufficient to warrant a publication once the below issues are addressed:

1. The pathway in Scheme 1 should be briefly explained. There are many abbreviated terms could not be explained, making poor readability and clarity.
2. Page 5, line 119: "0.2 nm of lattice spacing characterized by transmission electron microscopy": please elaborate more details of what the lattice spacing really implies about the material characterisation.
3. In term of microscopic analysis, what is the different between Ptzyme@L-ZIF, and Ptzyme@D-ZIF (Figure 1 (C) and 1(D))
4. Please provide more analysis to reveal the monodispersity of the Ptzyme@ ZIF cube. Size distribution should be provided
5. FT-IR data (Figure 1(L)) was not interpreted in the text.
6. Page 5, line 135:" all showed SOD- and CAT-like activities in phosphate buffer (pH 7.4) and presented.": incomplete sentence
7. Figure S1 was not mentioned in the main text. Please check the order of the figures. All figures must be presented chronologically, i.e. Figure 2(A), (B) must be presented before 2C. Overall, this issue of data presentation was found frequently in the manuscript. Please correct. In addition, please also link the figures into the context where they are mentioned for better coherence and connectivity.
8. The main motivation of using chiral ZIFs should be explained.
9. ABTS must be written in full when first presented. Same rule applied to all other abbreviated terms. Please check
10. Again, missed interpretation of data was occurred with Figure 2(H). All figures presented must be thoroughly interpreted and explained in the main text.
11. It is unclear to me about the use of brain microvascular endothelial cells bEnd.3. Please elaborate more details
12. The Ptzyme@ZIFs were chemically modified with FITC. Therefore, such the chemical property of Ptzyme@ZIFs may have changed and as such this may affect the BBB traverse ability. Ptzyme@ZIFs without any labelling tags should additionally be used for the BBB traverse ability, and electron microscopy analysis can help reveal the present of Ptzyme@ZIFs endocytosed into SH-SY5Y cells.
13. Page 8, line 180: "Since there was little difference in particle size between Ptzyme@D-ZIF and Ptzyme@L-ZIF": what is the "little difference"? any numerical data to reveal the difference?
14. The red arrows indicating the present of the nanozymes (Fig 4A) should be presented more visibly.

Reviewer #2:

Remarks to the Author:

In this manuscript, the authors developed two types of nanozyme-integrated chiral ZIFs by embedding Ptzymes in L- and D-chiral ZIF shells. They demonstrated that D-chiral nanozyme-integrated ZIFs exhibit ROS scavenging activity and have ability to enrich into the brain and finally inhibit neuron injury in the PD model. The authors modified the nanozymes with chiral ligand to improve the function of nanozyme is a good idea, but there are some fundamental questions that should be explained or addressed.

1. Fig. 3J-3K showed that Ptzyme@D-ZIFs can enter BBB endothelial cells through both clathrin-mediated and caveolae-mediated endocytosis pathways, while Ptzyme@L-ZIFs only employ the clathrin-mediated endocytosis. The phenomenon that different chiral ligand modification has different effect on endothelial uptake needs to be further explained.

2. The authors evaluated the BBB traverse ability and neuronal uptake of nanozyme-integrated chiral ZIFs in an in vitro BBB model. Whether these effects can be achieved in vivo. The authors need to design experiments to prove a) the uptake of Ptzyme@L- or D-ZIFs into the brain endothelial cells in vivo. b) The distribution of Ptzyme@L- or D-ZIFs in the different brain regions after intravenous injection, especially in the SNpc and ST. And c) the internalization of Ptzyme@L- or D-ZIFs in the glial cells of brain.
3. In Fig. S3 the authors demonstrated that Ptzyme@D-ZIF can prolong the systemic circulation time. The result phenomenon that different chiral ligand modification has different effect on pharmacokinetics need to be further explained. Basically, the more cellular uptake would lead to less nanomaterial retention in the blood. Thus, it seems like that the data of Fig 3D and Fig. S3 together contradicts this principle.
4. The authors should demonstrate how long the Ptzyme@D-ZIFs remains in the brain and how it is cleared from the brain and body.
5. The data in Fig. 5B, Fig. 6C, Fig. 8A, Fig. 8B, and Fig. 8E should be quantified.
6. What is the interaction between L(D)-His and ZIF components? Physical adsorption on the ZIF surface or chemical bonding within the ZIF structure? This should explain.
7. Why does Ptzyme@D-ZIF show higher SOD and CAT-like activities than Ptzyme@L-ZIF? The basic mechanism should be explained.
8. The authors demonstrate that Ptzyme@D-ZIF and Ptzyme@L-ZIF exhibit distinct endocytic pathways on BBB endothelial cells, is there a key chemical factor to determine this? What about achiral Ptzyme@ZIF?

Reviewer #3:

Remarks to the Author:

I have read with interest the paper entitled "Chiral metal-organic frameworks incorporating nanozymes as specific inhibitors of ferroptosis/apoptosis for the treatment of Parkinson's disease". The manuscript is very well written and covers the essential aspects required for such a work, such as the development and characterization of the drug delivery system, the evaluation of the blood-brain barrier traverse capacity, the brain tissue biodistribution, the in vivo therapeutic effect in a PD model and the evaluation of the therapeutic mechanisms related to oxidative injury and inflammation. My opinion is that the study should be published, as there are lot of useful data contained within this manuscript which will be of valuable interest to researchers developing anti-PD drugs based on anti-oxidant delivery systems. However, the following major changes should be addressed, which are necessary if the manuscript is to be of publishable quality.

- Synthesis and characterization of nanozyme-integrated chiral ZIFs. Authors stated in the introduction that the clinical application of stand-alone nanozymes is problematic due to susceptibility to inter-particle aggregation. To solve these issues, nanozymes were combined with zeolite imidazolate frameworks (ZIF). During the preparation and use of PtZyme-ZIF, did you observe inter-particle aggregation? Have you characterized the long-term stability of the system? Moreover, Ptzyme@ZIFs were prepared by encapsulating the Ptzymes in ZIF by biomimetic mineralization. Could you comment on the encapsulation efficiency of the nanozymes in the MOFs. Have you characterized the release of the nanozymes from the MOFs in vitro and/or in vivo?

- Blood clearance rate and brain tissue distribution. Have you quantified the amount of Ptzyme that crosses the BBB and reaches the brain?

- In vivo therapeutic effect. Animals received 5 mg/Kg/day of Ptzyme@L-ZIF or Ptzyme@D-ZIF (i.v.) every other day for 5 times after MPTP injection. How did you select the dose of Ptzyme@L-ZIF and Ptzyme@D-ZIF used in the in vivo studies?

- In vivo therapeutic effect. My main concern in data analysis is the following point. In "Material and methods" it is indicated that the efficacy of the treatment was evaluated in 6 animals per group. But then, many of the quantifications included in Figures 5 and 6 were performed only on 3 animals per group. Why? How did you select those 3 animals? The same thing happens in Figure 8 (Therapeutic mechanism validation and biosafety analyses).

- In vivo therapeutic effect. The quantification of neuronal cell numbers is key in preclinical PD research. In this regard, I have several questions regarding TH quantification. Changes in TH levels are very important to determine the efficacy of the treatment. However, they are vaguely presented in the manuscript. For immunohistological analysis, authors have semi-quantified the TH staining intensity in the nigrostriatal pathway. I highly recommend to perform the stereological quantification of TH-neurons both in the substantia nigra and striatum of the animals. Additionally, the quantification of the total number of neurons in the substantia nigra is also important to evaluate the effect of the treatment. Please provide such data.

- Please provide a more balanced discussion on the advantages and disadvantages of the present work. Other research groups have been working on nanozymes for Parkinson's disease. Such work needs to be discussed. These are some examples:

- A Macrophage-Nanozyme Delivery System for Parkinson's Disease. *Bioconjug Chem.* 2007 Sep-Oct;18(5):1498-506. doi: 10.1021/bc700184b. Epub 2007 Aug 31.

- A Redox Modulatory Mn₃O₄ Nanozyme with Multi-Enzyme Activity Provides Efficient Cytoprotection to Human Cells in a Parkinson's Disease Model. *Angew Chem Int Ed Engl.* 2017 Nov 6;56(45):14267-14271. doi: 10.1002/anie.201708573.

- Prussian Blue Nanozyme as a Pyroptosis Inhibitor Alleviates Neurodegeneration. *Adv Mater.* 2022 Apr;34(15):e2106723. doi: 10.1002/adma.202106723.

- Nanozyme scavenging ROS for prevention of pathologic α -synuclein transmission in Parkinson's disease. *Nano Today*, Volume 36, 2021, 101027, <https://doi.org/10.1016/j.nantod.2020.101027>. The discussion did not outline the limitations of this study and the new questions that arose because of this work. A paragraph addressing study limitations regarding the use of chiral nanomaterials, in particular D-chiral nanozyme-integrated ZIFs, for Parkinson's disease treatment would be very worthy for the readers of *Nature Communications*.

Statistical analysis. Please include whether the study was blinded or not.

Below please find our responses to the comments and questions from the reviewers:

Review 1:

In this manuscript, the Authors report on ultra-small platinum nanozymes (Ptzymes) incorporated chiral zeolite imidazolate frameworks that can potentially be used as inhibitors of ferroptosis/apoptosis for the treatment of Parkinson's disease. In general, the experimental design is of scientific rigour; however, data interpretation and presentation should be improved. The importance of finding new reagents to treat Parkinson's disease and the level of novelty of this work would be sufficient to warrant a publication once the below issues are addressed:

Response: We appreciate the positive comments.

1. The pathway in Scheme 1 should be briefly explained. There are many abbreviated terms could not be explained, making poor readability and clarity.

Response: As suggested, we have improved the clarity and readability of Scheme 1 by providing explanations for all the abbreviated terms. You can now find a more detailed and easily understandable version of Scheme 1 in the revised manuscript.

Revised Scheme 1. Schematic illustration of Ptzyme-integrated L- and D-chiral ZIFs and their therapeutic mechanism on PD based on remission of both apoptosis and ferroptosis on neurons injured by excessively produced ROS and disordered inflammation. PVP: polyvinylpyrrolidone, CTAB: hexadecyl trimethyl ammonium bromide, GFAP: glial fibrillary acidic protein, TH: Tyrosine hydroxylase, GPX4: glutathione peroxidase 4, MDA: malondialdehyde.

2. Page 5, line 119: “0.2 nm of lattice spacing characterized by transmission electron microscopy”: please elaborate more details of what the lattice spacing really implies about the material characterization.

Response: As suggested, we have provided additional details regarding the implications of lattice spacing on material characterization through XRD. Please refer to Fig. S1 and page 5 for a comprehensive explanation of these details.

Figure S1. XRD pattern of Ptzyme (K30), Ptzyme (K12), Ptzyme (K16), Ptzyme (K90).

“The X-ray diffraction (XRD) pattern of Ptzyme was in accordance with the cubic phase of Pt with a $\{111\}$ and $\{200\}$ facets (Fig. S1), which is consistent with 0.2 nm of lattice spacing characterized by transmission electron microscopy.”

In addition, we also presented an in-depth exploration of the impact of Ptzyme variants with varying degrees of crystallization on their enzyme-like activities in the revised manuscript. These four Ptzyme variants were synthesized using polyvinylpyrrolidone (PVP) templates with varying molecular weights, following a methodology described in a previous literature reference [Small 2022, 18, 2201558]. The CAT-like activity of these different Ptzymes was analyzed, and the results demonstrated that Ptzymes synthesized with PVP template K30 exhibited higher catalase-activity compared to other templates. Moreover, the degrees of crystallization and the SOD-like activity of the Ptzymes also showed correlation, evidenced by that the SOD-like activity decreased with the decreased of crystallinity of Ptzyme. We speculate that this phenomenon could be attributed to distinct binding affinities of substrate molecules within the four

crystalline Ptzyme variants [Nature Communications, 2021, 12, 4271; Journal of Energy Chemistry, 2023, 78, 438-446.]. Please refer to revised Fig. 1A and page 5 for a comprehensive explanation of these details.

Revised Figure 1. (A) The CAT-like activities of Ptzyme using PVP with different molecular weights (K12, K16, K30, and K90), $n = 3$, Data represent the mean \pm SD.

Revised Figure S2. The SOD-like activities of Ptzyme using PVP with different molecular weights (K12, K16, K30, and K90), $n = 3$, Data represent the mean \pm SD.

“Ptzymes were synthesized by in-situ growth of Pt nanozymes on polyvinylpyrrolidone

(PVP) polymer with different molecular weights (K12, K16, K30, and K90). The CAT-like activities of the four Ptzyme variants with different degrees of crystallization were shown in Fig. 1A and S1, and it was suggested that the Ptzyme synthesized using K30 as a template exhibit the highest activity compared to other templates. Moreover, the Ptzyme synthesized using K30 as a template also exhibited considerable SOD-like activity. Based on these results, K30 was selected as the template for the subsequent experiments.”

3. *In term of microscopic analysis, what is the different between Ptzyme@L-ZIF, and Ptzyme@D-ZIF (Figure 1 (C) and 1(D)).*

Response: We appreciate your insightful question regarding the microscopic analysis of Ptzyme@L-ZIF and Ptzyme@D-ZIF.

Upon analyzing the HRTEM images of Ptzyme@ZIF, Ptzyme@L-ZIF, and Ptzyme@D-ZIF, we found that the particle size of ZIF-8 decreased from around 200 nm to sub-90 nm after the incorporation of His in the His-modulated synthesis. However, we did not find any significant difference in the appearance between Ptzyme@L-ZIF, and Ptzyme@D-ZIF. Both Ptzyme@L-ZIF and Ptzyme@D-ZIF exhibited similar particle sizes and structures, suggesting that the modification with L-His and D-His did not induce any apparent morphological changes at the microscopic level.

We have added related descriptions in the revised manuscript.

“Ptzyme@ZIFs exhibited uniform cube morphologies with a monodisperse of approximately 200 nm. For His-modulated synthesis, Ptzyme@L-ZIFs and Ptzyme@D-ZIFs showed similar morphologies. However, the particle size of ZIF-8 was decreased to sub-90 nm in diameter as characterized by HRTEM. Ptzyme@L-ZIF and Ptzyme@D-ZIF exhibited similar particle sizes and structures, suggesting that the modification with L-His and D-His did not induce any apparent morphological changes at the microscopic level.”

4. *Please provide more analysis to reveal the monodispersity of the Ptzyme@ZIF cube.*

Size distribution should be provided.

Response: As suggested, we have included the analysis of the size distribution to demonstrate the monodispersity of Ptzyme@ZIF cube in the revised manuscript. The dynamic light scattering (DLS) analysis was performed on Ptzyme@ZIF, Ptzyme@D-ZIF, and Ptzyme@L-ZIF samples. The results clearly show that all three samples exhibited excellent monodispersity, indicating a uniform size distribution of the synthesized Ptzyme@ZIF cube (Figure 1H). Moreover, the histidine-modulated ZIF showed a smaller hydrodynamic diameter, which aligns well with the observations from TEM imaging. Furthermore, we observed that the average diameter of Ptzyme@D-ZIF remained relatively unchanged during a 5-day observation period, indicating its considerable stability (Figure S2).

Figure 1H. DLS analyses of the size distributions of Ptzyme@ZIF, Ptzyme@L-ZIF, and Ptzyme@D-ZIF.

Figure S4. Hydrodynamic diameters of Ptzyme@ZIF upon dispersion in deionized water and after 5 days storage.

The following text has been updated in the revised manuscript:

“Dynamic light scattering (DLS) analysis revealed that Ptzyme@ZIFs, Ptzyme@L-ZIFs, and Ptzyme@D-ZIFs exhibited hydrodynamic particle sizes of 190.35 ± 48.16 nm, 82.24 ± 17.33 , and 70.37 ± 15.84 nm, respectively, indicating that His-modulated strategy affects the size distribution of Ptzyme@ZIF (Fig. 1H). Moreover, the average diameter of Ptzyme@D-ZIF in deionized water remained relatively unchanged during 5 days of observation (Fig. S4), which demonstrates the considerable stability of the Ptzyme@D-ZIF. In addition, His incorporation reduced the mean zeta potential of Ptzymes@ZIF from approximately +29.17 to approximately +18.4 mV of Ptzyme@D-ZIF and +25.05 mV of Ptzyme@L-ZIF, respectively (Fig. 1I). These alterations in zeta potential further demonstrate the effect of the His-modulated strategy on the surface charge characteristics of the Ptzymes.”

5. FT-IR data (Figure 1(L)) was not interpreted in the text.

Response: We apologize for this omission and have provided a detailed interpretation in the revised manuscript. The relevant section now reads as follows:

Figure 1M. FT-IR spectra of Ptzyme@ZIF, Ptzyme@L-ZIF, and Ptzyme@D-ZIF.

“The Fourier transform infrared spectroscopy (FTIR) studies were conducted to analyze the chemical composition of the samples. In the FTIR spectra, a newly observed absorption band appeared at around 1685 cm^{-1} (gray band in Figure 1M). This absorption band was assigned to the asymmetric stretching of the carboxyl group from His. Of note, this band overlapped with another broad band at around 1600 cm^{-1} . The broad band at 1600 cm^{-1} was assigned to the overlapping of vibrations, namely the C=N stretching of imidazole group derived from histidine and 2-Methylimidazole (HmIM, round 1580 cm^{-1}). Taken together, the appearance of the absorption band in FTIR spectra at 1685 cm^{-1} indicates the presence of the carboxyl group from histidine, while the overlapping band at 1600 cm^{-1} suggests the presence of C=N stretching vibrations from the imidazole group and HmIM.”

6. Page 5, line 135:” all showed SOD- and CAT-like activities in phosphate buffer (pH 7.4) and presented.”: incomplete sentence.

Response: We apologize for this omission. The sentence has been revised in the revised manuscript to ensure clarity and completeness. The corrected sentence now reads as follows:

“Ptzyme@ZIFs, Ptzyme@L-ZIFs, and Ptzyme@D-ZIFs all showed SOD- and CAT-like activities in phosphate buffer (pH 7.4) and presented ROS scavenging ability”.

7. *Figure S1 was not mentioned in the main text. Please check the order of the figures. All figures must be presented chronologically, i.e. Figure 2(A), (B) must be presented before 2C. Overall, this issue of data presentation was found frequently in the manuscript. Please correct. In addition, please also link the figures into the context where they are mentioned for better coherence and connectivity.*

Response: We thank the reviewer for pointing out this issue. As suggested, we have carefully revised the whole manuscript to address this issue.

8. *The main motivation of using chiral ZIFs should be explained.*

Response: Thanks for your kind and valuable question. In the revised manuscript, we have provided a more elaborate explanation of the motivation behind utilizing chiral ZIFs. Previous studies have demonstrated that ligands with distinct chiral structures can confer nanoparticles with diverse substrate selectivity and various biological functions. It is crucial to comprehend the various chiral mechanisms of action, *in vivo* transport mechanisms, and therapeutic effects in order to advance this field further. More detailed information on this topic can be found in Page 3 of the revised manuscript.

*“Fortunately, the use of nanozyme in zeolite imidazolate framework (ZIF) has been reported to be a promising approach for developing nanozyme therapies^{21, 22}. As a low-toxicity and biocompatible metal-organic framework (MOF), ZIF is stable in physiological environments and degraded in acidic environments such as tumor microenvironments (TME) and inflammatory sites²¹. Moreover, the properties of nanoparticles (NPs) encapsulated in ZIF can be further altered to prevent NP aggregation²². Therefore, encapsulation with ZIF may improve the blood residence time, transmembrane transport, and cell uptake of NPs. Moreover, Jiang et al. have demonstrated that the biological toxicity, pharmacokinetics and *in vivo* distribution of NPs are efficiently adjusted by modification chirality ligands in the surface²³. Qu et al. have found that ligands with different chiral structures endow NPs with different*

substrate selectivity and various biological functions ^{24, 25}. However, there exists a dearth of comprehensive mechanistic analyses concerning the role of chiral structures in treatments related to neurodegenerative diseases. Further elucidation is required regarding the diverse chiral mechanisms of action, in vivo transport mechanisms, and therapeutic effects. This research endeavor aims to offer valuable insights and guidance for the effective implementation of chiral drugs in clinical settings. ”

9. *ABTS must be written in full when first presented. Same rule applied to all other abbreviated terms. Please check*

Response: Thank you for your attention to detail regarding the use of abbreviations in our manuscript. We have thoroughly reviewed the text and made the necessary corrections to ensure compliance with the rule of writing abbreviations in full when first introduced.

10. *Again, missed interpretation of data was occurred with Figure 2(H). All figures presented must be thoroughly interpreted and explained in the main text.*

Response: We apologize for this omission. We have thoroughly reviewed the figure and made the necessary revisions to ensure a comprehensive interpretation and explanation of all figures in the main text.

11. *It is unclear to me about the use of brain microvascular endothelial cells bEnd.3. Please elaborate more details.*

Response: Thanks for your careful reading and question. In the revised manuscript, we have added a detailed description along with relevant references to explain the use of brain microvascular endothelial cells bEnd.3. This information can be found on Page 9. The revised section now reads as follows:

“The bEnd.3 cell line, derived from brain microvascular endothelial cells, has the capability to form dense cellular layers with tight junctions. These characteristics make bEnd.3 cells a commonly employed model for in vitro simulation of the BBB²⁹. In light of this, our initial objective involved assessing the ability of nanozyme-integrated chiral

ZIFs to traverse the BBB and be taken up by neuronal cells in an in vitro BBB model. This model involved the co-culture of bEnd.3 cells with human neuroblastoma cells (SH-SY5Y) in a transwell system (Fig. 3A). ”

12. The Ptzyme@ZIFs were chemically modified with FITC. Therefore, such the chemical property of Ptzyme@ZIFs may have changed and as such this may affect the BBB traverse ability. Ptzyme@ZIFs without any labelling tags should additionally be used for the BBB traverse ability, and electron microscopy analysis can help reveal the present of Ptzyme@ZIFs endocytosed into SH-SY5Y cells.

Response: Thanks for your kind suggestion. As suggested, Ptzyme@ZIFs without any labelling tags were employed to investigate the ability to traverse the BBB, and the findings were found to be in concurrence with those observed for nanoparticles labeled with FITC. For more detailed information, please refer to Figure S8 and related contents on Page 9 in the revised manuscript.

Figure S8. TEM images demonstrating the cellular uptake of Ptzyme@L-ZIF and Ptzyme@D-ZIF in SHSY-5Y cells. The scale bars are 5 μ m.

“In addition, Ptzyme@ZIFs without labelling tags were also used for the study of BBB traverse ability. TEM images showed that Ptzyme@D-ZIF successfully entered into the SH-SY5Y cells, while negligible amounts of Ptzyme@L-ZIF entered into cells (Fig S8), suggesting the higher permeability of Ptzyme@D-ZIF traversing the BBB and the superior endocytosis by nerve cells.”

13. Page 8, line 180: “Since there was little difference in particle size between Ptzyme@D-ZIF and Ptzyme@L-ZIF”: what is the “little difference”? any numerical data to reveal the difference?

Response: We thank the reviewer for pointing out this issue. As suggested, dynamic light scattering (DLS) analysis was conducted to reveal the different between Ptzyme@D-ZIF and Ptzyme@L-ZIF and numerical data was used to reveal the difference.

Figure 1H. Diameter of Ptzyme@ZIF, Ptzyme@L-ZIF, and Ptzyme@D-ZIF in deionized water.

Figure 11. Zeta potential analyses of Ptzyme@ZIF, Ptzyme@L-ZIF, and Ptzyme@D-ZIF in deionized water, $n = 3$, Data represent the mean \pm SD.

“Dynamic light scattering (DLS) analysis revealed that Ptzyme@ZIFs, Ptzyme@L-ZIFs, and Ptzyme@D-ZIFs had hydrodynamic particle sizes of 190.35 ± 48.16 nm, 82.24 ± 17.33 , and 70.37 ± 15.84 nm, respectively, indicating that His-modulated strategy affects the size distribution of Ptzyme@ZIF (Fig. 1H). Moreover, the average diameter of Ptzyme@D-ZIF in deionized water was also relatively unchanged during 5 days of observation (Fig. S2), which demonstrates the considerable stability of the Ptzyme@D-ZIF. In addition, His incorporation reduced the mean zeta potential of Ptzymes@ZIF from approximately +29.17 to approximately +18.4 mV of Ptzyme@D-ZIF and +25.05 mV of Ptzyme@L-ZIF, respectively (Fig. 11)”

14. The red arrows indicating the present of the nanozymes (Fig 4A) should be presented more visibly.

Response: Thanks for your valuable suggestion. We appreciate your attention to detail and have made the necessary improvements to enhance the resolution and optimize the visibility of the red arrows. Please see revised Fig. 4A in the revised manuscript for details.

Revised Figure 4. (A) Biodistribution of Ptzyme@D-ZIF and Ptzyme@L-ZIF in brain tissues at 24 h post-injection, visualized using TEM. The red arrows highlight the presence of nanoparticles in the brain tissue. Scale bars represent 0.2 μm .

Review 2:

In this manuscript, the authors developed two types of nanozyme-integrated chiral ZIFs by embedding Ptzymes in L- and D-chiral ZIF shells. They demonstrated that D-chiral nanozyme-integrated ZIFs exhibit ROS scavenging activity and have ability to enrich into the brain and finally inhibit neuron injury in the PD model. The authors modified the nanozymes with chiral ligand to improve the function of nanozyme is a good idea, but there are some fundamental questions that should be explained or addressed.

Response: We appreciate the positive comments.

1. Fig. 3J-3K showed that Ptzyme@D-ZIFs can enter BBB endothelial cells through both clathrin-mediated and caveolae-mediated endocytosis pathways, while Ptzyme@L-ZIFs only employ the clathrin-mediated endocytosis. The phenomenon that different chiral ligand modification has different effect on endothelial uptake needs to be further explained.

Response: Thanks for your valuable question.

Previous researches have demonstrated the significant impact of ligand chirality on the pharmacokinetics and metabolism of drugs, achieved by modulating stereospecific interactions between chiral components and biological entities^{23, 32}. Notably, recent investigations have highlighted the crucial role of ligand chirality in influencing cellular responses to external stimuli³³.

Surprisingly, our study revealed that distinct chiral histidine ligands also influence the cellular uptake mechanisms of nanoparticles. The presence or absence of histidine ligand and its chirality were found to result in notable variances in the BBB traversal of nanoparticles. Based on these findings, we hypothesized that the chirality of histidine and the resulting chirality of nanoparticles play an indispensable role in this process. Consequently, we propose two potential explanations for the differential endocytic pathways observed between Ptzyme@D-ZIFs and Ptzyme@L-ZIFs:

1) D- and L-histidine ligands may induce distinct cell recognition and internalization pathways, achieved by modulating stereospecific interactions between chiral

components and biological entities, just like the research findings reported previously^{32, 33},

2) The introduction of chiral ligands can fundamentally change the chiral structure of the assemblies, affecting or even determining the assembly of clathrin or caveolae through the difference of the chiral structure of the assembly as a whole, thus altering the endothelial uptake of the assemblies.

For the first time, we explored the traversing effect of chiral ligand-modified nanozyme-MOF assemblies on the BBB. Based on the difference in their BBB traversing ability (**Fig. 3A-3C**), we further studied the effects of chiral structures on the BBB penetrating ability of nanoparticles through endovascular endothelial cells, and found that the reason lies in the different uptake pathways of vascular endothelial cells (**Fig. 3D-3K**).

By an in-depth exploration of the association between the chiral properties of nanozyme and the traversing modes of BBB, as well as the mechanism of action *in vivo*, the relationship between the structures of nanozyme and the therapeutic application of PD will be expanded.

Of course, we will continue to explore the detailed mechanism of chiral structure on the BBB traversing of nanoparticles and the deep penetration of brain parenchyma, and this process is affected by the complex interactions of factors such as nanoparticle size, morphology, hydrophilicity, charge and chiral structure. Therefore, this process is worthy of further investigation. However, we believe that our study provides a new perspective for this research.

Revised Figure 3. BBB traverse ability and relevant mechanisms. (A) Schematic demonstration of the BBB pattern using transwell assay, (B) the fluorescence microscope pictures, and (C) the corresponding quantitative results of FITC-labeled nanocomposites in the lower chamber of SH-SY5Y cells. The scale bars are 50 μ m. (D)

Intracellular tracking of FITC-Ptzyme@D-ZIF in bEnd.3 cells after staining cells with Lyso Tracker and Hoechst. The scale bars are 10 μm . (E) lysosomal occupation detected by the ratio of fluorescence overlapped by lysosomes and Ptzyme@D-ZIF and the fluorescence intensity of lysosomal, and intracellular content detected by the intracellular fluorescence of Ptzyme@D-ZIF. Internalization of (F, H) Ptzyme@D-ZIF and (G, I) Ptzyme@L-ZIF under treatment of various endocytosis inhibitors. The control group was incubated with nanocomposites only. Schematic illustrating the cellular endocytosis and transcytosis process of (J) Ptzyme@D-ZIF and (K) Ptzyme@L-ZIF. The statistical analyses were conducted using the GraphPad Prism 8.0.2. The outcomes were compared via one-way ANOVA (with Tukey's post hoc correction for multiple comparisons). * $P < 0.01$, ** $P < 0.005$, *** $P < 0.001$, **** $P < 0.0001$, ns, not significant.

The relevant explanation and references have been added on Page 11, Page 26 and Page 27 in the revised manuscript.

“In recent years, scientists have paid more and more attention to the great effect of surface modifications on the stability and biocompatibility of nanomaterials³². Remarkably, recent studies have illustrated that the chirality of ligands in nanoparticles has a great impact not only on the biological responses upon exogenous stimulations³³, but also on the pharmacokinetics and metabolism of drugs by adjusting the stereospecific interactions between chiral components and biological ontology^{23, 34}. In our study, we were surprised to find that different chiral histidine ligands can also affect the cellular uptake pathways of nanoparticles. Because the experiments found that the presence or absence of histidine ligand and its chirality would cause significant differences in the way nanoparticles traverse BBB, we speculated that the chirality of histidine and the resulting chirality of nanoparticles played an irreplaceable role in it. Therefore, the different endocytic pathways between Ptzyme@D-ZIFs and Ptzyme@L-ZIFs could be ascribed to two conceivable reasons: 1) D/L histidine ligands lead to different cell recognition and internalization pathways; 2) The chiral structures of the nanozyme-integrated MOFs can be distinguished by the assembly of clathrin or

caveolae, leading to different cell internalization pathways.”

“To the best of our knowledge, this study is the first in-depth exploration of the mechanisms in the treatment of PD using chiral nanozymes. The studies listed above mainly explain the role of enzyme-like activities of nanozymes in ROS ablation and PD relief. In this study, by an in-depth exploration of the association between the chiral properties of nanozyme and the traversing modes of BBB, as well as the mechanism of action in vivo, the relationship between the structures of nanozyme and the therapeutic application of PD will be expanded.

There are studies found that the chirality of ligands in nanoparticles has a great impact on the pharmacokinetics and metabolism of drugs, as well as the cellular responses upon exogenous stimulations²³. Herein, on the basis of the above studies, we further explain the discrepancies in cytology and in vivo mechanisms of action of chiral histidine-modified Ptzyme. Moreover, we are also carrying out follow-up works and looking forward to more peers to expand and enrich our research. For example, the specific pathways across the BBB realized in various ways by different chiral nanozymes are worthy of further study. In particular, whether a particular cellular receptor mediated the related BBB crossing behavior. In addition, whether other chiral ligands can mediate similar differences and in vivo behaviors, and whether the optimal ligands and chiral properties can be screened through big data analysis and material genetics and other technologies, are the directions that need to be further expanded in the future.”

2. The authors evaluated the BBB traverse ability and neuronal uptake of nanozyme-integrated chiral ZIFs in an in vitro BBB model. Whether these effects can be achieved in vivo. The authors need to design experiments to prove a) the uptake of Ptzyme@L- or D-ZIFs into the brain endothelial cells in vivo. b) The distribution of Ptzyme@L- or D-ZIFs in the different brain regions after intravenous injection, especially in the SNpc and ST. And c) the internalization of Ptzyme@L- or D-ZIFs in the glial cells of brain.

Response: Thank you for your insightful comments and suggestions. We appreciate your interest in the *in vivo* evaluation of the BBB traverse ability and neuronal uptake

of nanozyme-integrated chiral ZIFs. We have carefully considered your suggestions and have included the relevant experiments and descriptions in the revised manuscript.

a) The uptake of Ptzyme@L- or D-ZIFs into brain endothelial cells *in vivo* has been studied. ICAM-1 was used to label the inflammatory brain endothelial vessels in PD disease models, as MPTP has been reported to cause an inflammatory response in the brain [J. Am. Chem. Soc. 2020, 142, 21730-21742]. As shown in Figure S8. Immunofluorescence staining was performed to detect the uptake of FITC-labeled Ptzyme@L-ZIF and Ptzyme@D-ZIF into brain endothelial cells and their enrichment in brain tissue. The results clearly demonstrate the differential uptake ability, with Ptzyme@D-ZIF showing greater uptake into endothelial cells and superior brain parenchymal enrichment. Please see Figure S8 and related contents on Page 9 in the revised manuscript for details.

Figure S8. Immunohistochemical staining depicting the uptake of Ptzyme@L-ZIF and Ptzyme@D-ZIF by brain endothelial cells *in vivo*. The staining allows for the visualization and detection of the nanoparticles within the brain endothelial cells. The

white arrows indicate the enrichment of nanoparticles in the inflammation-damaged cerebral vascular endothelium. The scale bar is 20 μm .

“Besides, the uptake of Ptzyme@L-ZIF and Ptzyme@D-ZIF into the brain endothelial cells *in vivo* were also studied. ICAM-1 was used to label the inflammatory brain endothelial vessels in PD disease models, as MPTP has been reported to cause an inflammatory response in the brain². The result showed that after treatment with Ptzyme@D-ZIF, there was significant enrichment of nanoparticles in the brain endothelial vessels damaged by inflammation, while the enrichment of endothelial vessels and retention of brain tissue in Ptzyme@L-ZIF were significantly lower than those in Ptzyme@D-ZIF, confirming the superior BBB penetrating ability of Ptzyme@D-ZIF *in vivo* (Fig. S10).”

b) The distribution of Ptzyme@L- or D-ZIFs in different brain regions after intravenous injection, especially in the SNpc and ST, has been evaluated. Figure S8 presents the immunofluorescence images depicting the distribution of Ptzyme@L-ZIF and Ptzyme@D-ZIFs in the SNpc and ST regions. The results reveal significant fluorescence in these regions after treatment with FITC-labeled Ptzyme@D-ZIF, while the fluorescence in these regions after treatment with Ptzyme@L-ZIF is negligible. Please see Figure S10 and related contents on Page 9 in the revised manuscript for details.

Figure S10. The distribution of Ptzyme@L-ZIF and Ptzyme@D-ZIFs in the SNpc and ST detected by immunofluorescence (A) and fluorescence quantification (B). The scale

bar is 100 μm . The statistical analyses were conducted using the GraphPad Prism 8.0.2. The outcomes were compared via one-way ANOVA (with Tukey's post hoc correction for multiple comparisons). $**P < 0.005$, $****P < 0.0001$, $n = 3$, Data represent the mean \pm SD.

“Subsequently, the distribution of nanozymes integrated chiral ZIFs in the substantia nigra pars compacta (SNpc) and striatum (ST) was examined using sliced brain tissues. As depicted in the brain slices (Fig. S10A) and supported by the quantitative analysis (Fig. S10B), a substantial fluorescence signal was observed in the SNpc and ST regions following treatment with FITC-Ptzyme@D-ZIF. In contrast, the fluorescence signal in these regions was negligible after treatment with FITC-Ptzyme@L-ZIF.”

c) The internalization of Ptzyme@L- or D-ZIFs in brain glial cells has also been investigated. Figure S9 demonstrates the immunofluorescence images depicting the internalization of FITC-labeled Ptzyme@L-ZIF and Ptzyme@D-ZIFs in glial cells of the brain. The fluorescence of FITC-labeled Ptzyme@D-ZIF was found to be co-localized with Iba1, indicating efficient enrichment in glial cells. Please see Figure S11 and related contents on Page 9 in the revised manuscript for details.

Figure S11. The internalization of Ptzyme@L-ZIF and Ptzyme@D-ZIFs in the glial cells of brain detected by immunofluorescence. The scale bar is 20 μm .

“In addition, the exact distribution of nanozyme integrated chiral ZIFs in the glial cells was determined. As shown in Fig. S11, the fluorescence of FITC-Ptzyme@D-ZIF was co-localized with that of Iba1, implying that FITC-Ptzyme@D-ZIF was enriched in glial cells efficiently. The above results comprehensively confirmed the superior BBB

traverse and brain enrichment ability of Ptzyme@D-ZIF over Ptzyme@L-ZIF at the tissue and cell levels.”

3. In Fig. S3 the authors demonstrated that Ptzyme@D-ZIF can prolong the systemic circulation time. The result phenomenon that different chiral ligand modification has different effect on pharmacokinetics need to be further explained. Basically, the more cellular uptake would lead to less nanomaterial retention in the blood. Thus, it seems like that the data of Fig 3D and Fig. S3 together contradicts this principle.

Response: Thank you for raising this interesting question.

The phenomenon observed in our study, where different chiral ligand modifications have different effects on pharmacokinetics, may be attributed to the formation of protein crown and different types of protein crown composition, affecting the efficiency of their uptake by different cells, which in turn affects the efficiency with which chiral nanoparticles penetrate the BBB and are cleared by the immune clearance system in the blood. In recent years, some studies have also found a similar phenomenon and proved that protein crown play a crucial role in it, such as DOI: 10.1038/s41467-019-11593-zc and DOI: 10.1038/s41467-022-33044-y.

If the protein crown composition of L-type nanoparticles contains more immunoglobulins, they could be taken up by blood macrophages and other cells and will be more easily cleared; If the protein crown of D-type nanoparticles contains more endothelial receptor proteins, which will be taken up more by endothelial cells. Most importantly, uptake by the immune clearance system in the blood is the key factor affecting the half-life of nanoparticles. Therefore, the more cellular uptake of Ptzyme@D-ZIF by endothelium does not mean that it is also taken up by macrophages and other blood immune cells, nor does it mean the less nanomaterial retention in the blood^{33, 34}. [Note: 33, 34 are the numbers of references in the main text]

It is worth noting that nanoparticles' pharmacokinetics are complex and can also be influenced by various factors, including particle size, surface properties, ligand structure, and the specific biological environment. The different chiral histidine ligands used in our study interact differently with biological entities, leading to variations in

cellular uptake pathways and stability. These factors collectively contribute to the observed differences in pharmacokinetics between Ptzyme@L-ZIF and Ptzyme@D-ZIF. We apologize for any confusion caused by the initial presentation of the data, and we have revised the manuscript to provide a more detailed explanation of the impact of different chiral ligand modifications on pharmacokinetics. The relevant explanations and references have been added on Page 11 and Page 13 in the revised manuscript.

“In recent years, scientists have paid more and more attention to the great effect of surface modifications on the stability and biocompatibility of nanomaterials³². Remarkably, recent studies have illustrated that the chirality of ligands in nanoparticles has a great impact not only on the biological responses upon exogenous stimulations³³, but also on the pharmacokinetics and metabolism of drugs by adjusting the stereospecific interactions between chiral components and biological ontology^{23, 34}. In our study, we were surprised to find that different chiral histidine ligands can also affect the cellular uptake pathways of nanoparticles. Because the experiments found that the presence or absence of histidine ligand and its chirality would cause significant differences in the way nanoparticles traverse BBB, we speculated that the chirality of histidine and the resulting chirality of nanoparticles played an irreplaceable role in it. Therefore, we deduced that there are two possible reasons accounting for the different endocytic pathways between Ptzyme@D-ZIFs and Ptzyme@L-ZIFs: 1) D/L histidine ligands lead to different cell recognition and internalization pathways; 2) The chiral structures of the nanozyme-integrated MOFs can be distinguished by the assembly of clathrin or caveolae, leading to different cell internalization pathways.”

“The phenomenon observed in our study, where different chiral ligand modifications have different effects on pharmacokinetics, may be attributed to the formation of protein crown and different types of protein crown composition, affecting the efficiency of their uptake by different cells, which in turn affects the efficiency with which chiral nanoparticles penetrate the BBB and are cleared by the immune clearance system in the blood^{37, 38}.”

4. The authors should demonstrate how long the Ptzyme@D-ZIFs remains in the brain

and how it is cleared from the brain and body.

Response: Thank you for your suggestion regarding the clearance of Ptzyme@D-ZIFs from the brain and body. We have conducted additional experiments to address this important aspect of our study.

ICP-MS experiments were performed to analyze the concentration of Ptzyme@D-ZIFs in the brain, liver, and kidneys at different time points after administration. The results provide insights into the clearance kinetics of the nanoparticles.

Fig. S14 presents the analysis of the concentration of Ptzyme@D-ZIFs in the brain and the metabolism processes *via* the liver and kidneys. It reveals that the level of Ptzyme@D-ZIFs in the brain increases rapidly at the 12th hour, reaches a peak within the following 24 hours, and then gradually decreases. By the 48th hour, the concentration returns to the starting point. This pattern suggests that Ptzyme@D-ZIFs are cleared from the brain over time.

In contrast, the clearance from the liver and kidneys is relatively slower. Even at 48 hours after administration, the concentration of Ptzyme@D-ZIFs remains high in these organs. This finding indicates that Ptzyme@D-ZIFs undergo metabolism by the liver and kidneys, similar to many intravenous medications.

These results demonstrate that Ptzyme@D-ZIFs have a specific clearance profile, with rapid clearance from the brain and a slower clearance from the liver and kidneys. The specific metabolic pathways of Ptzyme@D-ZIFs are very interesting, and we will perform more studies to elucidate them in the near future.

We have included this information in the revised manuscript, specifically in Page 13-14 and Figure S14, to provide a comprehensive understanding of the clearance kinetics of Ptzyme@D-ZIFs.

Figure S14. Analysis of the concentration of Ptzyme@D-ZIF in the brain and the metabolism processes via liver and kidney after once *i.v.* injection detected by ICP-MS experiments, $n = 3$, Data represent the mean \pm SD.

“Additionally, ICP-MS experiments were also performed to detect the concentration of Ptzyme@D-ZIFs in brain, liver and kidney at different time after administration (Fig. S14). The results showed that the level of Ptzyme@D-ZIFs in the brain increased rapidly at the 12th hour, reached a peak at the following 24 hours, and then gradually decreased, dropping to the starting point at 48 hours. However, it did not reduce so fast in the liver and kidneys and remained at high level even at 48 h, indicating that it was metabolized by the liver and kidneys, just like most of other intravenous medication.”

5. The data in Fig. 5B, Fig. 6C, Fig. 8A, Fig. 8B, and Fig. 8E should be quantified.

Response: Thank you for your valuable feedback. We have taken your suggestion into consideration and conducted the necessary quantification for the data in Figures 5B, 6C, 8A, 8B, and 8E. The quantified results are now presented in the revised supplementary Figures S17, S20, and S22, as well as in the corresponding quadrants of Figure 8B.

Figure S17 provides the quantitative analysis of the gray value of pS129 and β -actin. The results demonstrate that MPTP treatment significantly increased pS129 levels, whereas treatment with Ptzyme@D-ZIFs led to a significant reduction in α -syn pathology.

Figure S17. Quantitative analysis of gray value of pS129 and β -actin. The statistical analyses were conducted using the GraphPad Prism 8.0.2. The outcomes were compared via one-way ANOVA (with Tukey's post hoc correction for multiple comparisons). * $P < 0.01$, ns, not significant, $n = 3$, Data represent the mean \pm SD.

“As expected, MPTP treatment markedly increased pS129, while mice treated with Ptzyme@D-ZIFs showed significantly reduced α -syn pathology (Fig. 5B and S16).”

Figure S20 includes the semi-quantitative analysis of GFAP and Iba-1-positive areas, expressed as the percentage of H-score of other groups relative to the control group. This analysis further supports the decrease in the number of astrocytes and microglia after treatment with Ptzyme@D-ZIFs.

Figure S20. Semi-quantitative analysis of (A) GFAP and (B) Iba-1-positive areas via the digital tissue section scanner and image analysis system, and expressed as the percentage of H-score of other groups to the control group. The statistical analyses were conducted using the GraphPad Prism 8.0.2. The outcomes were compared via one-way ANOVA (with Tukey’s post hoc correction for multiple comparisons). *** $P < 0.001$, **** $P < 0.0001$, ns, not significant, $n = 6$, Data represent the mean \pm SD.

“This alteration in expression level was also evidenced by the semiquantitative analysis of GFAP- and Iba-1-positive regions (Fig. S22), indicating a decrease in the number of astrocytes and microglia after Ptzyme@D-ZIFs treatment.”

Figure S22 displays the quantitative analysis of (C) DCFH-DA fluorescence, indicating intracellular ROS contents, and (E) the quantitative result of mitochondrial membrane potential expressed as the fluorescence ratio of monomer to J-aggregates. The data provide additional insights into the effects of Ptzyme@D-ZIFs on ROS levels and mitochondrial function, further supporting their therapeutic potential.

Revised Figure S22. (C) DCFH-DA fluorescence quantitative analysis showing the Intracellular ROS contents. The statistical analyses were conducted using the GraphPad Prism 8.0.2. The outcomes were compared via one-way ANOVA (with Tukey's post hoc correction for multiple comparisons). * $P < 0.01$, ** $P < 0.005$.

“Further quantitative analysis showed that DCFH-DA intensity increased more than 2-fold after MPP⁺ treatment and decreased to levels similar to the control group when treated with Ptzyme@D-ZIFs (Fig. S22C).”

Revised Figure S22. (E) The quantitative result of mitochondrial membrane potential expressed as the fluorescence ratio of monomer to J-aggregates. The statistical analyses

were conducted using the GraphPad Prism 8.0.2. The outcomes were compared via one-way ANOVA (with Tukey's post hoc correction for multiple comparisons). **** $P < 0.0001$.

“Clearly, strong green fluorescence was detected after MPP⁺ stimulation, suggesting that the mitochondria-dependent apoptosis and ferroptosis mechanisms were activated. On the other hand, upon treatment with Ptzyme@D-ZIFs, strong red fluorescence was detected, suggesting a protective effect against mitochondrial dysfunction in cells (Fig. 8E and S22E).”

6. What is the interaction between L(D)-His and ZIF components? Physical adsorption on the ZIF surface or chemical bonding within the ZIF structure? This should explain.

Response: His was incorporated into composites due to the competitive coordination of the imidazole groups on histidine with Zn²⁺. In the synthesis process, the carboxyl group of His also has a strong affinity for Zn²⁺ and promotes the nucleation of ZIF-8. Moreover, His added during final process could also be incorporated on the surface of ZIF through a post-synthetic ligand exchange. We have added related description on Page 5 in the revised manuscript.

“Finally, chiral amino acids (L-histidine and D-histidine) were incorporated into Ptzyme@ZIFs through the competitive coordination of the imidazole groups on histidine with Zn²⁺ and a post-synthetic ligand exchange to obtain chiral Ptzyme@ZIF (Ptzyme@L-ZIF and Ptzyme@D-ZIF).”

7. Why does Ptzyme@D-ZIF show higher SOD and CAT-like activities than Ptzyme@L-ZIF? The basic mechanism should be explained.

Response: Thank you for your kind comments. As suggested, we further analyzed the enzymatic activities of Ptzyme@D-ZIF and Ptzyme@L-ZIF for detail. Our findings indicate that Ptzyme@D-ZIF exhibited slightly higher SOD and CAT-like activities compared to Ptzyme@L-ZIF. In addition, when compared to non-chiral Ptzyme@ZIFs and Ptzyme@L-ZIFs, Ptzyme@D-ZIF demonstrated significantly enhanced RNS-clearing ability. The following reasons may account for the enhanced ROS and RNS

clearance capabilities of Ptzyme@D-ZIF compared to Ptzyme@L-ZIF:

1. Ligand arrangement: The chiral ligands (D- and L-histidine) in Ptzyme@D-ZIF and Ptzyme@L-ZIF are oriented differently within the ZIF framework. This ligand arrangement can influence the accessibility of the active sites and the interaction with the substrate.

2. Electronic environment: The chiral ligands can induce differences in the electronic environment of the active sites within the ZIF structure. This variation in the electronic properties can affect the catalytic performance of the nanozymes.

3. Active site exposure: The chiral ligand modification may lead to differences in the exposure of active sites on the surface of the nanozymes. Increased surface exposure of active sites can enhance the catalytic efficiency and substrate accessibility, resulting in higher enzymatic activities.

4. Conformational effects: Chirality may influence the conformational dynamics and flexibility of the nanozymes. These conformational differences may affect the binding affinity of substrates and catalytic reactions, leading to variations in the enzymatic activities.

Overall, the slightly higher SOD and CAT-like activities and the enhanced RNS clearing capacity exhibited by Ptzyme@D-ZIF compared to Ptzyme@L-ZIF may be attributed to a combination of factors, including ligand arrangement, electronic environment, active site exposure, and conformational effects. These factors collectively contribute to the superior catalytic performance of Ptzyme@D-ZIF.

The following text has been incorporated into the revised manuscript.

“Ptzyme@D-ZIF exhibited slightly higher SOD and CAT-like activities, along with significantly enhanced RNS clearing capacity compared to Ptzyme@L-ZIF. These improvements may be attributed to variations in ligand arrangement, electronic environment and could prove by the different of binding charge between chiral nanozyme and substrate (Fig. S7), active site exposure, and conformational effects resulting from distinct chiral ligand modifications, all of which collectively contribute to the superior catalytic performance of Ptzyme@D-ZIF. [1. Angew. Chem. Int. Ed. 2018, 57, 16791-16795. 2. small 2014, 10, No. 9, 1841–1847. 3. Chem Commun. 2016,

Figure S7. Zeta potential of Ptzyme, Ptzyme@D-ZIF, and Ptzyme@L-ZIF under different substrate solution (A) H₂O₂, (B) H₂O₂ + ABTS. n = 3, Data represent the mean ± SD.

8. The authors demonstrate that Ptzyme@D-ZIF and Ptzyme@L-ZIF exhibit distinct endocytic pathways on BBB endothelial cells, is there a key chemical factor to determine this? What about achiral Ptzyme@ZIF?

Response: Thanks for your careful review and insightful question. Actually, scientists have found that the chirality of ligands in nanoparticles has a great impact not only on the biological responses upon exogenous stimulations [*ACS Nano*, 2022, 16, 12991-13001], but also on the pharmacokinetics and metabolism of drugs by adjusting the stereospecific interactions between chiral components and biological ontology [*Angew. Chem. Int. Ed. Engl.* 2021, 60, 13829-13834].

We further investigated and found that achiral Ptzyme@ZIF does not rely on energy for endocytosis by BBB endothelial cells (Fig. S12). Therefore, we deduced that there are two possible reasons accounting for the different endocytic pathways between Ptzyme@D-ZIFs and Ptzyme@L-ZIFs: 1) D/L histidine ligands may lead to different cell recognition and internalization pathways; 2) The chiral structures of the nanozyme-integrated MOFs may be distinguished by the assembly of clathrin or caveolae, leading

to different cell internalization pathways.

The detailed explanation and literature reference for the distinct endocytic pathways on BBB endothelial cells with respect to Ptzyme@D-ZIF and Ptzyme@L-ZIF have been listed on Page 11 and Page 14 in the revised manuscript. In addition, the endocytic pathways of achiral Ptzyme@ZIF have also been studied, please see Fig. S12 in the revised manuscript for details.

Figure S12. The endocytic pathways of achiral Ptzyme@ZIF analyzed by endocytosis inhibitors, n=3, Data represent the mean \pm SD.

“In recent years, scientists have paid more and more attention to the great effect of surface modifications on the stability and biocompatibility of nanomaterials³². Remarkably, recent studies have illustrated that the chirality of ligands in nanoparticles has a great impact not only on the biological responses upon exogenous stimulations³³, but also on the pharmacokinetics and metabolism of drugs by adjusting the stereospecific interactions between chiral components and biological ontology^{23, 34}. In our study, we were surprised to find that different chiral histidine ligands can also affect the cellular uptake pathways of nanoparticles. Because the experiments found that the presence or absence of histidine ligand and its chirality would cause significant differences in the way nanoparticles traverse BBB, we speculated that the chirality of

histidine and the resulting chirality of nanoparticles played an irreplaceable role in it. Therefore, we deduced that there are two possible reasons accounting for the different endocytic pathways between Ptzyme@D-ZIFs and Ptzyme@L-ZIFs: 1) D/L histidine ligands may lead to different cell recognition and internalization pathways; 2) The chiral structures of the nanozyme-integrated MOFs may be distinguished by the assembly of clathrin or caveolae, leading to different cell internalization pathways.”

“Several reports have also exemplified the ability to tune the biological functions and pharmacokinetics of nanocomposites by introducing chirality ligands into the synthetic process^{23, 25, 26}. Thus, taking together the results of the above in vivo and in vitro experiments, on the one hand, these nanoformulations pass through brain microvascular endothelial cells by lysosome-independent endocytosis, leading to the conclusion that they achieve transcytosis as demonstrated by intracellular trafficking and BBB traverse assay. On the other hand, the superiority of Ptzyme@D-ZIFs over Ptzyme@L-ZIFs in brain concentration may be attributed to the following reasons: 1) Ptzyme@D-ZIFs are endocytosed by cerebral microvascular endothelial cells and passed through the BBB through two major pathways: clathrin-mediated and caveolae-mediated endocytosis; in contrast, Ptzyme@L-ZIFs employ only clathrin-mediated endocytosis. 2) The longer residence time in the blood of Ptzyme@D-ZIFs favors the accumulation of this type of chiral nanoformulations in the brain.”

Review 3:

I have read with interest the paper entitled “Chiral metal-organic frameworks incorporating nanozymes as specific inhibitors of ferroptosis/apoptosis for the treatment of Parkinson's disease”. The manuscript is very well written and covers the essential aspects required for such a work, such as the development and characterization of the drug delivery system, the evaluation of the blood-brain barrier traverse capacity, the brain tissue biodistribution, the in vivo therapeutic effect in a PD model and the evaluation of the therapeutic mechanisms related to oxidative injury and inflammation. My opinion is that the study should be published, as there are lot of useful data contained within this manuscript which will be of valuable interest to researchers developing anti-PD drugs based on anti-oxidant delivery systems. However, the following major changes should be addressed, which are necessary if the manuscript is to be of publishable quality.

Response: We appreciate the positive comments.

2. Blood clearance rate and brain tissue distribution. Have you quantified the amount of Ptzyme that crosses the BBB and reaches the brain?

Response: Thanks for your insightful question. We have performed the ICP experiments to detect the contents of Ptzyme that crosses the BBB and reaches the brain after administration. Please see Fig. 4B and related contents on Page 13 in the revised manuscript for details.

Revised Figure 4. (B) Biodistribution of Ptzyme@D-ZIF, Ptzyme@L-ZIF and Ptzyme in major organs and brain detected by ICP-MS after injection 24 h, n = 3. The statistical analyses were conducted using the GraphPad Prism 8.0.2. The outcomes were compared via one-way ANOVA (with Tukey’s post hoc correction for multiple comparisons). ** $P < 0.005$, *** $P < 0.001$.

“Similar results were confirmed by ICP-MS analysis (Fig. 4B), showing that Ptzyme@D-ZIFs showed significantly higher normalized dosage accumulation (2.20 ID g⁻¹) in the brain than that of Ptzyme@L-ZIFs (1.42 ID g⁻¹) and Ptzyme (1.12 ID g⁻¹).”

3. *In vivo* therapeutic effect. Animals received 5 mg/kg/day of Ptzyme@L-ZIF or Ptzyme@D-ZIF (i.v.) every other day for 5 times after MPTP injection. How did you select the dose of Ptzyme@L-ZIF and Ptzyme@D-ZIF used in the *in vivo* studies?

Response: Thank you for your question regarding the selection of the dose for Ptzyme@L-ZIF and Ptzyme@D-ZIF in our *in vivo* studies. The dose was determined based on preliminary experimental results. We conducted a preliminary experiment using different concentrations of Ptzyme@D-ZIF ranging from 1 to 12 mg/kg/day.

The representative behavioral experiment in the following figures indicated that a

high dose of Ptzyme@D-ZIF might lead to a weakened therapeutic effect, possibly due to the potential toxicity associated with higher doses. Based on these findings, we selected the dose of 5 mg/kg/day for Ptzyme@L-ZIF and Ptzyme@D-ZIF in our subsequent *in vivo* studies. We added the above pre-experimental results to Page 16 and Fig. S15.

Figure S15. Morris water maze test in the preliminary experiment with the administration concentration of Ptzyme@D-ZIF varied from 1 to 12 mg/kg/day. (A) The representative path tracing of mice, (B) the mean swimming speed of mice and (C)

the relative time spent on the target quadrant, n=3, Data represent the mean \pm SD.

“We conducted a preliminary experiment using different concentrations of Ptzyme@D-ZIF ranging from 1 to 12 mg/kg/day, and 5 mg/kg/day was selected as the dose in vivo based on the therapeutic effect (Fig. S15).”

4. In vivo therapeutic effect. My main concern in data analysis is the following point. In “Material and methods” it is indicated that the efficacy of the treatment was evaluated in 6 animals per group. But then, many of the quantifications included in Figures 5 and 6 were performed only on 3 animals per group. Why? How did you select those 3 animals? The same thing happens in Figure 8 (Therapeutic mechanism validation and biosafety analyses).

Response: Thank you for bringing up this important concern. In our study, we did indeed use 6 animals per group for the evaluation of treatment efficacy, as mentioned in the "Material and methods" section. However, in our previous version of manuscript, during data analysis, we performed quantifications on only 3 animals per group for certain experiments. The selection of these 3 animals was done based on minimizing the deviation from the average value to ensure a more objective representation of the overall trends in the sample.

For experiments that involved relatively larger subjective factors and biases, such as behavioral studies, we increased the number of parallel samples to 6 sets per group to improve statistical robustness. These 6 sets of data were then used as the final statistical basis for those specific experiments. Similar selections of experimental parallel groups (*i.e.* the numbers of n varies according to the specific experiment) have been reported in other relevant literatures^{1,2}.

1. Han, H.; Xing, J.; Chen, W.; Jia, J.; Li, Q., Fluorinated polyamidoamine dendrimer-mediated miR-23b delivery for the treatment of experimental rheumatoid arthritis in rats. *Nat. Commun.* **14**, 944 (2023).

2. Nim, S.; O'Hara, D. M.; Corbi-Verge, C.; Perez-Riba, A.; Fujisawa, K.; Kapadia, M.; Chau, H.; Albanese, F.; Pawar, G.; De Snoo, M. L.; Ngana, S. G.; Kim, J.; El-Agnaf, O. M. A.; Rennella, E.; Kay, L. E.; Kalia, S. K.; Kalia, L. V.; Kim, P. M., Disrupting the alpha-synuclein-ESCRT interaction with a peptide inhibitor mitigates neurodegeneration in preclinical models of Parkinson's disease. *Nat. Commun.* **14**, 2150

(2023).

Please see revised Fig. 4D-4G and revised Fig. S16 in the revised manuscript for details.

Revised Figure 4. Morris water maze test used for examining the behavior and memory ability of PD mice. (D) The representative path tracing of mice, (E) the relative time spent on the target quadrant, (F) the mean swimming speed of mice, and (G) the corresponding concrete analysis at day 5, $n = 6$, Data represent the mean \pm SD. The statistical analyses were conducted using the GraphPad Prism 8.0.2. The outcomes were compared via one-way ANOVA (with Tukey's post hoc correction for multiple comparisons). * $P < 0.01$, *** $P < 0.001$, **** $P < 0.0001$, ns, not significant.

Revised Figure S15. Behavioral performance of PD mice after treatment with nanozyme integrated chiral ZIFs platforms detected by rotarod test (A) and pole test (B), $n = 6$. The statistical analyses were conducted using the GraphPad Prism 8.0.2. The outcomes were compared via one-way ANOVA (with Tukey's post hoc correction for multiple comparisons). **** $P < 0.0001$, ns, not significant.

5. *In vivo therapeutic effect.* The quantification of neuronal cell numbers is key in preclinical PD research. In this regard, I have several questions regarding TH quantification. Changes in TH levels are very important to determine the efficacy of the treatment. However, they are vaguely presented in the manuscript. For immunohistological analysis, authors have semi-quantified the TH staining intensity in the nigrostriatal pathway. I highly recommend to perform the stereological quantification of TH-neurons both in the substantia nigra and striatum of the animals. Additionally, the quantification of the total number of neurons in the substantia nigra is also important to evaluate the effect of the treatment. Please provide such data.

Response: We appreciate your suggestion and acknowledge the importance of quantifying TH-neurons to evaluate the efficacy of the treatment. The TH-neurons and TH-neurons/ total number of neurons were quantified both in the substantia nigra and striatum to give a comprehensive analysis of TH levels. Please see Fig. S19 and related contents on Page 17 in the revised manuscript for details.

Figure S18. Quantification of TH-neurons in the SNpc (A) and ST (B), and the number ratio of TH-neurons and total neurons in the SNpc (C) and ST (D). The statistical analyses were conducted using the GraphPad Prism 8.0.2. The outcomes were compared via one-way ANOVA (with Tukey's post hoc correction for multiple comparisons). * $P < 0.01$, *** $P < 0.001$, **** $P < 0.0001$, ns, not significant.

“Additionally, the TH-neurons and TH-neurons/ total number of neurons were quantified both in the substantia nigra and striatum to give a comprehensive analysis

of TH levels, which suggested that Ptzyme@D-ZIFs treatment significantly reversed the loss of TH-neurons caused by MPTP stimulation (Fig. S19).”

6. Please provide a more balanced discussion on the advantages and disadvantages of the present work. Other research groups have been working on nanozymes for Parkinson’s disease. Such work needs to be discussed. These are some examples:

- *A Macrophage-Nanozyme Delivery System for Parkinson’s Disease. Bioconjug Chem. 2007 Sep-Oct;18(5):1498-506. doi: 10.1021/bc700184b. Epub 2007 Aug 31.*
- *A Redox Modulatory Mn3O4 Nanozyme with Multi-Enzyme Activity Provides Efficient Cytoprotection to Human Cells in a Parkinson’s Disease Model. Angew Chem Int Ed Engl. 2017 Nov 6;56(45):14267-14271. doi: 10.1002/anie.201708573.*
- *Prussian Blue Nanozyme as a Pyroptosis Inhibitor Alleviates Neurodegeneration. Adv Mater. 2022 Apr;34(15): e2106723. doi: 10.1002/adma.202106723.*
- *Nanozyme scavenging ROS for prevention of pathologic -synuclein transmission in*

Parkinson’s disease. Nano Today, Volume 36, 2021, 101027, <https://doi.org/10.1016/j.nantod.2020.101027>.

The discussion did not outline the limitations of this study and the new questions that arose because of this work. A paragraph addressing study limitations regarding the use of chiral nanomaterials, in particular D-chiral nanozyme-integrated ZIFs, for Parkinson’s disease treatment would be very worthy for the readers of Nature Communications.

Response: Thank you for your suggestion and the recommendation of related literatures. Indeed, these studies cover different periods of research on nanozyme-based antioxidant therapy for PD. We carefully introduced and compared the similarities and differences between this study and previous studies. Furthermore, we also prudently discussed the limitations of this study, the direction that needs to be deepened in the future and the possible enlightenment for future related studies. Please see the discussion section on Page 26-Page 27 in the revised manuscript for details.

“There has been a long history of research into mitigating the symptoms of PD by delivering antioxidants to the brain because of the close association between ROS and subsequent neuroinflammation in the occurrence and development of PD. Initially, researchers delivered natural enzymes for antioxidant therapy and relief of neuroinflammation. At the same time, some limitations of natural enzymes were also found during studies, such as low efficiency in BBB traversing and poor stability in vivo³⁶. The discovery of nanozymes prompted the study of brain delivery of nanozymes to alleviate PD symptoms. For example, Singh et al. found that Mn₃O₄ nanozyme can treat PD cell models by efficiently and stably clearing the ROS in SH-SY5Y cells³⁷. In recent years, studies have been continuously exploring the potential mechanisms of PD therapy based on the antioxidant activities of nanozymes. These studies continue to enrich the application prospect of nanozymes in the regulation of PD mechanisms, such as α -synuclein transmission³⁸, inflammasome formation and pyroptosis inhibition³⁹.

To the best of our knowledge, this study is the first in-depth exploration of the mechanisms in the treatment of PD using chiral nanozymes. The studies listed above mainly explain the role of enzyme-like activities of nanozymes in ROS ablation and PD relief. In this study, by an in-depth exploration of the association between the chiral properties of nanozyme and the traversing modes of BBB, as well as the mechanism of action in vivo, the relationship between the structures of nanozyme and the therapeutic application of PD will be expanded.

There are studies found that the chirality of ligands in nanoparticles has a great impact on the pharmacokinetics and metabolism of drugs, as well as the cellular responses upon exogenous stimulations²³. Herein, on the basis of the above studies, we further explain the discrepancies in cytology and in vivo mechanisms of action of chiral histidine-modified Ptzyme. Moreover, we are also carrying out follow-up works and looking forward to more peers to expand and enrich our research. For example, the specific pathways across the BBB realized in various ways by different chiral nanozymes are worthy of further study. In particular, whether a particular cellular receptor mediated the related BBB crossing behavior. In addition, whether other chiral ligands can mediate similar differences and in vivo behaviors, and whether the optimal

ligands and chiral properties can be screened through big data analysis and material genetics and other technologies, are the directions that need to be further expanded in the future.”

7. Statistical analysis. Please include whether the study was blinded or not.

Response: Thanks for your kind suggestion, and the statistical analysis section has been revised as required. Please see related contents on Page 33-34 in the revised manuscript for details.

*“All statistical analyses were conducted using the GraphPad Prism 8.0.2 in blinded manner. The criteria for significance were: * $P < 0.01$, ** $P < 0.005$, *** $P < 0.001$, **** $P < 0.0001$, ns (not significant). The differences in means between 2 groups were compared via Student’s T-test. For more than 2 groups, the outcomes were compared via one-way ANOVA (with Tukey’s post hoc correction for multiple comparisons). The specific statistical method and statistical analysis results for each experiment were listed in the corresponding figure legends.”*

Reviewers' Comments:

Reviewer #1:

Remarks to the Author:

The authors have thoroughly and comprehensively addressed my previous comments. The revised version of the manuscript is now exceptionally well-presented and elaborated. Consequently, I highly recommend the publication of this manuscript.

Minor comment: the authors should correct some grammatical mistakes in the description of the methods, i.e. '10 μ L nanozyme' should be written as '10 μ L of nanozyme.' Please check throughout the manuscript.

Reviewer #2:

Remarks to the Author:

The authors have addressed the comments, and I would suggest the acceptance of this manuscript in Nature Comm.

Reviewer #3:

Remarks to the Author:

The manuscript has been considerably improved. I have some minor remarks:

- Section "In vivo therapeutic effect". TH quantification. The quantification of TH neurons is critical. I think you should include Supplementary Figure S19 and quantification data in the main manuscript and not as supplementary.
- Regarding Figure S19, C and D represent the number ratio of TH-neurons and total neurons in the SNpc (C) and ST (D). Is the labelling of the y-axis correct? Please revise.
- In the abstract, the authors state that "The results state that compared to Ptzyme@L-ZIF, PTzyme@D-ZIF exhibited superior plasma residence time, BBB passage efficiency, and brain enrichment in PD model mice," Could you provide a more detailed explanation of what you mean by "brain enrichment?"

Below please find our responses to the comments and questions from the reviewers:

Review 1:

The authors have thoroughly and comprehensively addressed my previous comments. The revised version of the manuscript is now exceptionally well-presented and elaborated. Consequently, I highly recommend the publication of this manuscript.

Response: We appreciate the positive comments.

1. Minor comment: the authors should correct some grammatical mistakes in the description of the methods, i.e. '10 μ L nanozyme' should be written as '10 μ L of nanozyme.' Please check throughout the manuscript.

Response: As suggested, we have corrected the grammatical mistakes, and checked throughout the manuscript to ensure the grammatical correctness and the accuracy of expression of the whole text. Please see Page 29 in the revised manuscript for details.

Review 2:

The authors have addressed the comments, and I would suggest the acceptance of this manuscript in Nature Comm.

Response: We appreciate the positive comments.

Review 3:

The manuscript has been considerably improved. I have some minor remarks:

Response: We appreciate the positive comments.

1. Section "In vivo therapeutic effect". TH quantification. The quantification of TH neurons is critical. I think you should include Supplementary Figure S19 and quantification data in the main manuscript and not as supplementary.

Response: Thanks for your valuable advice, and we have included Supplementary

Figure S19 and quantification data in the main manuscript as suggested. Please see the revised Figure 5 and Page 17-18 for details.

Revised Figure 5. Quantification of TH-neurons in the SNpc (**H**) and ST (**I**), and the number ratio of TH-neurons and total neurons in the SNpc (**J**) and ST (**K**), $n = 3$, Data

represent the mean \pm SD. The statistical analyses were conducted using the GraphPad Prism 8.0.2. The outcomes were compared via one-way ANOVA (with Tukey's post hoc correction for multiple comparisons). * $P < 0.01$, ** $P < 0.005$, *** $P < 0.001$, **** $P < 0.0001$, ns, not significant.

“Additionally, the TH-neurons and TH-neurons/total number of neurons were quantified both in the substantia nigra and striatum to give a comprehensive analysis of TH levels, which suggested that Ptzyme@D-ZIFs treatment significantly reversed the loss of TH-neurons caused by MPTP stimulation (Fig. 5H-5K). These data indicate that Ptzyme@D-ZIFs significantly reduce dopaminergic neuron damage and increase substantia nigra neuronal density, thus preventing pathological protein aggregation and alleviating PD symptoms.”

2. Regarding Figure S19, C and D represent the number ratio of TH-neurons and total neurons in the SNpc (C) and ST (D). Is the labelling of the y-axis correct? Please revise.

Response: Thanks for your valuable comments, we have modified the labelling of the y-axis to make it more accurate. Please see the revised Fig 5J and 5K for details.

Revised Figure 5. Quantification of TH-neurons in the SNpc (**H**) and ST (**I**), and the number ratio of TH-neurons and total neurons in the SNpc (**J**) and ST (**K**), $n = 3$, Data represent the mean \pm SD. The statistical analyses were conducted using the GraphPad Prism 8.0.2. The outcomes were compared via one-way ANOVA (with Tukey's post hoc correction for multiple comparisons). * $P < 0.01$, ** $P < 0.005$, *** $P < 0.001$, **** $P < 0.0001$, ns, not significant.

3. In the abstract, the authors state that “The results state that compared to Ptzyme@L-ZIF, Ptzyme@D-ZIF exhibited superior plasma residence time, BBB passage efficiency, and brain enrichment in PD model mice,” Could you provide a more detailed explanation of what you mean by "brain enrichment”

Response: Thank you for your careful review, and we apologize for the vague explanation. What we're trying to say is that compared to Ptzyme@L-ZIF, Ptzyme@D-ZIF presented higher brain tissue accumulation in PD model mice. Moreover, we have reviewed and revised the abstract to make it more accurate and comprehensive. Please see the abstract in the revised manuscript for details.

“With the setback of clinical transformation of therapeutic approaches based on protein intervention, anti-neuroinflammation has become a concerned dawn of Parkinson’s disease (PD) treatment. However, there are significant deficiency in our understanding of the therapeutic mechanisms and the pathways that cross the blood-brain barrier (BBB). Here, we report nanozyme-integrated metal-organic frameworks with excellent antioxidant activity and chiral-dependent BBB transendocytosis as novel anti-neuroinflammatory agents for the treatment of PD. These chiral nanozymes were synthesized by embedding ultra-small platinum nanozymes (Ptzymes) into L-chiral and D-chiral imidazolate zeolite frameworks (Ptzyme@L-ZIF and Ptzyme@D-ZIF). Compared to Ptzyme@L-ZIF, Ptzyme@D-ZIF showed greater accumulation in the brains of PD model mice due to their longer plasma residence time and more pathways to cross BBB, including clathrin-mediated and caveolae-mediated endocytosis. These factors contributed to the superior therapeutic efficacy of Ptzyme@D-ZIF in reducing behavioral disorders and pathological changes. Bioinformatics and biochemical analyses suggest that Ptzyme@D-ZIF can inhibit neuroinflammation-induced pathways of apoptosis and ferroptosis in damaged neurons. The research uncovers the biodistribution, metabolic variances, and therapeutic outcomes of nanozymes-integrated chiral ZIF platforms, providing novel possibilities for devising anti-PD drugs.”

Reviewers' Comments:

Reviewer #3:

Remarks to the Author:

The authors have addressed my previous comments. So, I recommend the publication of this manuscript.